# Large-scale electrophysiology and deep learning reveal distorted neural signal dynamics after hearing loss

**Shievanie Sabesan[1], Andreas Fragner[2], Ciaran Bench[1], Fotios Drakopoulos[1], Nicholas A Lesica[1]\***

[1]Ear Institute, University College London, London, United Kingdom; [2]Perceptual Technologies, London, United Kingdom

**Abstract** Listeners with hearing loss often struggle to understand speech in noise, even with a hearing aid. To better understand the auditory processing deficits that underlie this problem, we made large-scale brain recordings from gerbils, a common animal model for human hearing, while presenting a large database of speech and noise sounds. We first used manifold learning to identify the neural subspace in which speech is encoded and found that it is low-dimensional and that the dynamics within it are profoundly distorted by hearing loss. We then trained a deep neural network (DNN) to replicate the neural coding of speech with and without hearing loss and analyzed the underlying network dynamics. We found that hearing loss primarily impacts spectral processing, creating nonlinear distortions in cross-frequency interactions that result in a hypersensitivity to background noise that persists even after amplification with a hearing aid. Our results identify a new focus for efforts to design improved hearing aids and demonstrate the power of DNNs as a tool for the study of central brain structures.

## Editor's evaluation

This fundamental work uses deep neural networks to simulate activity evoked by a wide range of stimuli and demonstrates systematic differences in latent population representations between hearing-impaired and normal-hearing animals that are consistent with impaired representations of speech in noise. The evidence supporting the conclusions is compelling, and the neural-network approach is novel with potential future applications. The research will be of interest to auditory neuroscientists and computational scientists.

**\*For correspondence:**
n.lesica@ucl.ac.uk

## Introduction

Hearing loss is a widespread problem with far-reaching consequences ranging from lost productivity and social isolation to decreases in quality of life and mental health (*Wilson et al., 2017*). It also imposes a growing societal burden with associated costs approaching $1 trillion annually (*World Health Organization, 2021*). Hearing aids are the only widely available treatment for hearing loss, but, unfortunately, current devices provide limited benefit in many workplace and social settings (*Lesica, 2018*).

The term 'hearing loss' does not capture the full spectrum of the effects of cochlear damage on auditory processing. One common consequence of cochlear damage is a loss of sensitivity that renders low-intensity sounds inaudible. This is the 'hearing loss' that is assessed in standard clinical tests and is addressed through amplification with hearing aids. But cochlear damage has other consequences beyond lost sensitivity that cause many people to struggle with high-intensity sounds that

are well above their audibility threshold (*Moore, 2007*). For example, people with hearing loss often have difficulties understanding speech in noisy settings, with or without a hearing aid (*Larson et al., 2000*). The pathophysiology underlying these high-intensity deficits remains poorly understood, and, as a result, they are largely ignored by clinicians and hearing aid designers.

The consequences of cochlear damage for the processing of both low- and high-intensity sounds have been well described at the level of the auditory nerve (AN; *Young, 2008*). In addition to the general decrease in neural activity resulting from lost sensitivity, there are also complex changes in the spatiotemporal structure of the neural activity patterns that encode acoustic information, such as lost synchrony capture (*Miller et al., 1997*) and distorted tonotopy (*Henry et al., 2016*). Many theories have attempted to explain how the peripheral changes associated with hearing loss might lead to perceptual deficits (*Humes and Dubno, 2010*; *Plomp, 1986*). But, with few explicit studies comparing neural coding in central auditory areas before and after hearing loss, it has been difficult to differentiate between competing theories or to identify which peripheral changes are most important to address.

One recent study of individual neurons in the inferior colliculus (IC) with and without hearing loss and hearing aids found that some properties, such as phoneme selectivity, were impacted while others, such as frequency selectivity and trial-to-trial variability, were not (*Armstrong et al., 2022*). But, while characterizing the impact of hearing loss on neural coding in individual neurons may be sufficient at the level of the AN (a set of largely homogeneous fibers driven by a few thousand inner hair cells), neural coding in downstream areas such as the IC, which are much larger and more complex, likely involves emergent network-level properties that are not readily apparent in the activity of individual neurons.

One of the challenges in characterizing neural coding at the network level is the high dimensionality of the activity patterns. If the goal is to gain insight into the underlying computations that the activity reflects, it can be useful to find more compact representations of the full activity that retain its important features. This process, often termed 'manifold learning' (*Mitchell-Heggs et al., 2023*; *Williamson et al., 2019*), typically utilizes techniques such as principal component analysis (PCA) that identify a projection of the full activity into a lower dimensional space that retains as much of its variance as possible. In this study, we use manifold learning to investigate the impact of hearing loss on the neural coding of speech in gerbils, a common animal model for the study of human hearing. We employ large-scale intracranial recordings that allow us to achieve comprehensive sampling of activity from individual animals at the fine spatial and temporal scales that are critical for encoding speech (*Garcia-Lazaro et al., 2013*).

We begin with the traditional approach to manifold learning using PCA to identify and analyze the low-dimensional subspace in which most of the variance in the full network activity patterns resides. We focus on the signal manifold, which captures the features of neural activity that are sound-evoked, first establishing that our recordings are sufficient to identify the signal manifold in individual animals and then that the changes in signal dynamics with hearing loss are fundamental, that is, that the dynamics within the signal manifold are not simply attenuated by hearing loss, but instead are truly distorted. We then continue our analysis using deep neural networks (DNNs) to perform manifold learning within the framework of a stimulus encoding model, which allows us to investigate the impact of hearing loss on the coding of novel sounds.

We demonstrate that training DNNs on our recordings allows for accurate prediction of neural activity in conjunction with identification of the signal manifold. We use the trained DNNs to probe the processing of basic acoustic features and show that hearing loss predominantly affects spectral, rather than temporal, processing. We then probe the processing of speech and find that this impaired spectral processing creates a hypersensitivity to background noise that persists even with a hearing aid and appears to arise from aberrant cross-frequency interactions. Our results demonstrate the power of DNNs to provide new insights into neural coding at the network level and suggest that new approaches to hearing aid design are required to address the highly nonlinear nature of the effects of hearing loss on spectral processing.

## Results

We recorded neural activity from the IC of anesthetized gerbils using electrode arrays with a total of 512 channels (*Armstrong et al., 2022*), allowing us to sample widely from neurons that were sensitive

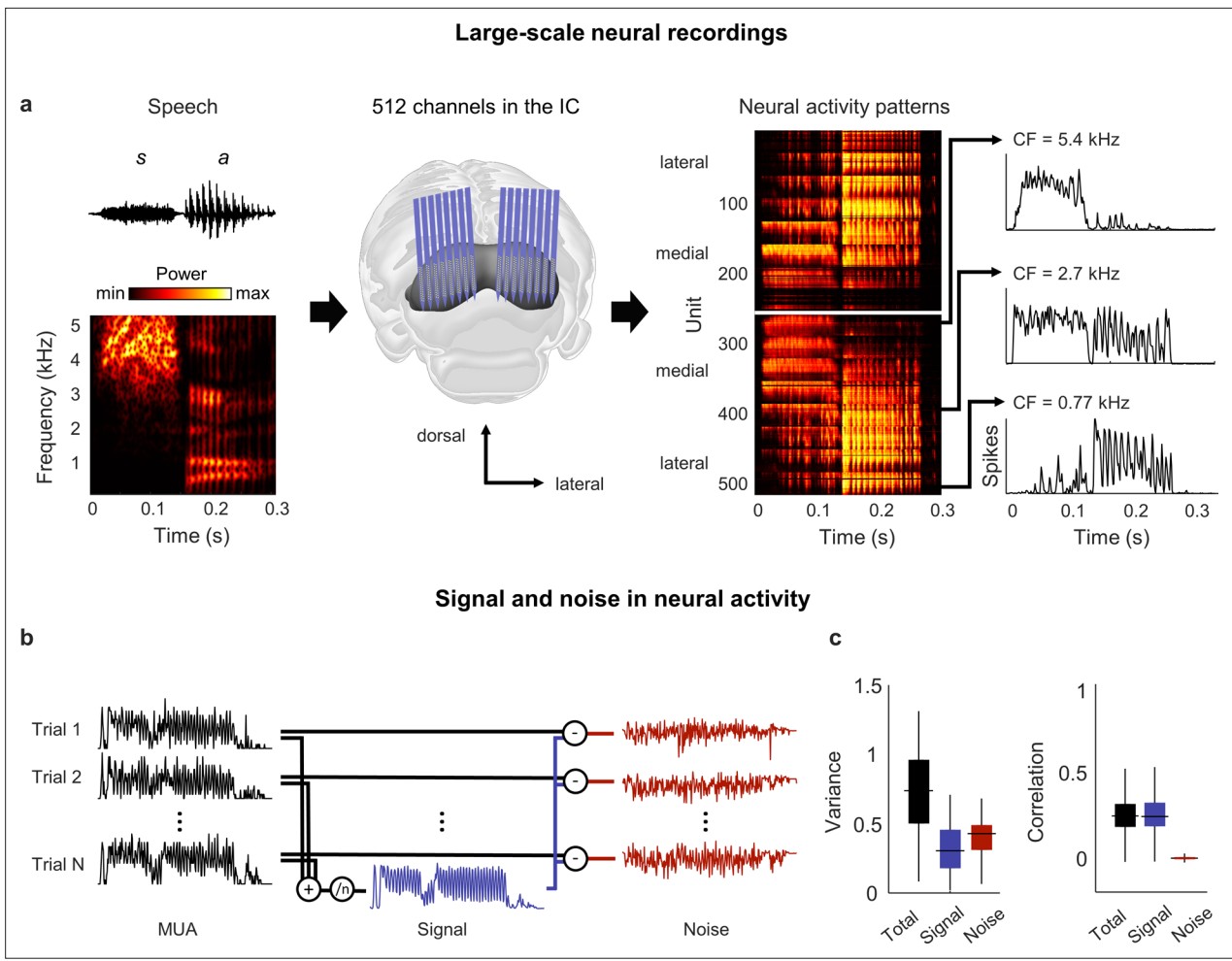

**Figure 1.** Neural signal and noise in the gerbil inferior colliculus (IC). (**a**) Schematic diagram showing the geometry of custom-designed electrode arrays for large-scale recordings in relation to the gerbil IC (center), along with the speech syllable 'sa' (left) and the neural activity that it elicited during an example recording (right). Each image of the neural activity corresponds to one hemisphere, with each row showing the average multi-unit activity recorded on one electrode over repeated presentations of the syllable, with the units arranged according to their location within the IC. The activity for three units with different center frequencies (CF; frequency for which sensitivity to pure tones is maximal) are shown in detail. (**b**) Schematic diagram showing the method for separating signal and noise in neural activity. The signal is obtained by averaging responses across repeated presentations of identical sounds. The noise is the residual activity that remains after subtracting the signal from the response to each individual presentation. (**c**) Signal and noise in neural activity. Left: total, signal, and noise variance in neural activity for units recorded from normal hearing animals (horizontal line indicates median, thick vertical line indicates 25th through 75th percentile, thin vertical line indicates 5th through 95th percentile; n = 2556). Right: total, signal, and noise correlation in neural activity for pairs of units recorded from normal hearing animals (n = 544,362).

to the full range of speech frequencies (*Figure 1a*). We recorded activity in response to more than 10 hr of speech in each animal, presenting the entire TIMIT speech database (*Garofolo, 1993*) twice – once in quiet and once in background noise, with the overall intensity, speech-to-noise ratio, and noise type varied from sentence to sentence (with a small number of sentences repeated multiple times under identical conditions to assess trial-to-trial variability). We processed the recordings to extract multi-unit activity (MUA) spike counts for each recording channel, using 1.3 ms time bins to account for the fact that neurons in the IC can encode information about speech with millisecond temporal precision (*Garcia-Lazaro et al., 2013*).

## The neural signal manifold is low dimensional

We began by analyzing activity from animals with normal hearing. As a first step toward characterizing the neural code for speech at the network level, we determined whether there was shared variance across units that would allow us to reduce the dimensionality of the activity patterns. Previous

work has shown that the correlations in IC activity are dominated by 'signal' (features of activity that are reproducible across repeated trials and, thus, convey acoustic information) rather than 'noise' (features of activity that vary from trial-to-trial and reflect intrinsic noise or fluctuations in brain state) (*Figure 1b*; *Garcia-Lazaro et al., 2013*).

Signal correlations were also dominant in our recordings: although signal variance (the covariance in activity across repeated trials) accounted for only 40% of the overall variance in activity, signal correlations accounted for 95% of the total correlation between units (*Figure 1c*). For a network operating in such a regime, with each neuron having more than half of its variance uncorrelated with that of its neighbors, there is limited scope for reducing the dimensionality of the full activity patterns. However, given the large signal correlations, it may be possible to identify a low-dimensional subspace in which the acoustic information represented by the signal is embedded.

We partitioned the recordings from each animal into two sets: a training set that was used to identify the principal components (PCs) of the activity (*Figure 2a*, step 1) and a test set with two repeated trials that was used to measure the variance that could be explained by each PC. To measure the total variance explained (*Figure 2a*, steps 2a–c), we projected the activity from test trial 1 onto the PCs, then reconstructed the original activity from the same trial using the PC projection and compared the reconstruction to the original activity. The overall dimensionality of the neural activity was high, as expected, with a large number of PCs required for the reconstruction to explain 95% of the total variance in the original activity (*Figure 2b*).

To measure the signal variance explained (*Figure 2a*, step 3), we compared the same reconstructed activity to the original activity from test trial 2 (when using activity from one trial to reconstruct activity on another, only those features that reliably encode acoustic information across trials can be successfully reconstructed and, thus, only signal variance can be explained). The signal variance explained saturated quickly, indicating that the signal dimensionality was much lower than the overall dimensionality, with only a small number of PCs (between 5 and 10) required to explain 95% of the signal variance in the original activity (*Figure 2c*).

These results suggest that the acoustic information in IC activity is restricted to a low-dimensional subspace, which we term the neural *signal manifold*, and that PCA is able to identify the dimensions that define this manifold. To confirm that PCA preferentially identified the signal manifold, we computed the fraction of the variance explained by each PC that was signal rather than noise (measured as the covariance between the activity on the two test trials when projected onto each PC relative to the overall variance after the same projection) and verified that it decreased with each successive PC (*Figure 2d*).

If the signal manifold reflects something fundamental about auditory processing, then we should expect the activity within it, which we term the *signal dynamics*, to be similar across animals with the same hearing status. To measure the similarity of the signal dynamics across animals (*Figure 2a*, step 4), we projected the original activity for each animal onto its respective signal manifold and then determined how much of the variance in the activity from one animal could be explained by the activity from another (allowing for additional linear transformation). We found that the signal dynamics for different animals were remarkably similar, with the signal dynamics from one animal accounting for, on average, 96% of the variance in the signal dynamics from other animals (*Figure 2e*). This result gives us confidence that the signal manifold is indeed fundamental, and that our methods are sufficient to identify it robustly in individual animals.

## Hearing loss distorts neural signal dynamics

We next sought to use analysis of the signal manifold to better understand the impact of hearing loss on the neural code for speech at the network level. We induced sloping mild-to-moderate sensorineural hearing loss (*Figure 2—figure supplement 1*) by exposing gerbils (n = 6) to broadband noise using established protocols (*Armstrong et al., 2022*; *Suberman et al., 2011*). After waiting at least 1 mo for the effects of the hearing loss to stabilize, we made neural recordings while presenting the same speech and noise sounds and then performed the same manifold learning.

The results for animals with hearing loss were similar to those for animals with normal hearing: the overall dimensionality of the neural activity was high (*Figure 2f*); the dimensionality of the signal manifold was low (*Figure 2g*; between 4 and 7 PCs required to explain 95% of the signal variance); and the signal dynamics were similar across animals (*Figure 2h*; 95% average variance explained),

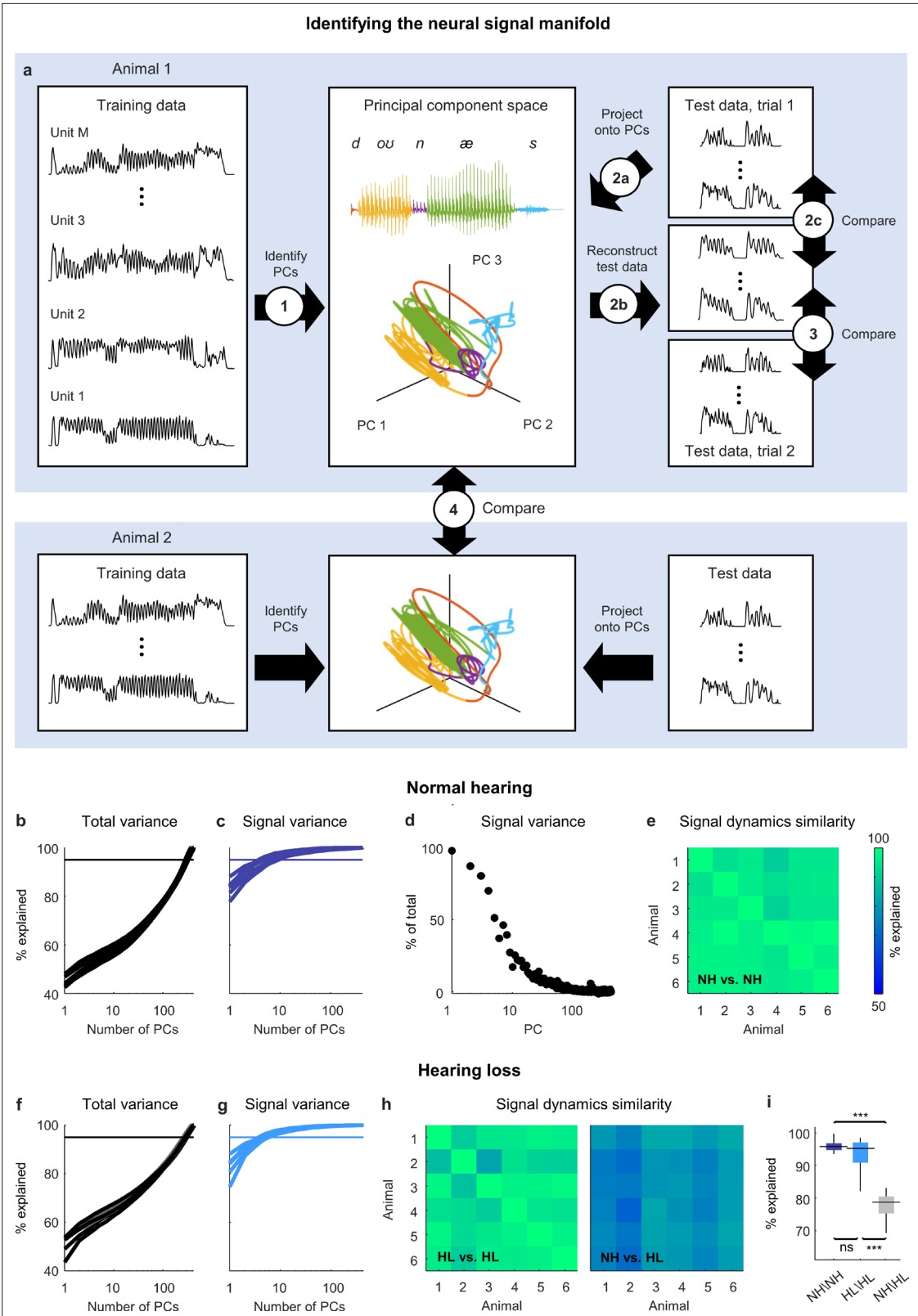

**Figure 2.** Neural signal dynamics identified via classical methods. (**a**) Schematic diagram showing the method for identifying the neural signal manifold from recordings of neural activity. Step 1: principal component analysis (PCA) is performed on a subset of the recordings allocated for training. Step 2a: a subset of recordings allocated for testing are projected onto the principal components (PCs) from step 1. Step 2b: The projections from step 2a are used to reconstruct the test recordings. Step 2c: the reconstructions from step 2b are compared to the test recordings from step 2a to determine the

*Figure 2 continued on next page*

*Figure 2 continued*

total variance explained. Step 3: the reconstructions from step 2b are compared to another set of test recordings (made during a second presentation of the same sounds) to determine the signal variance explained. Step 4: the projections from step 2a for one animal are compared to the projections for another animal to determine the similarity of the signal dynamics between animals. (**b**) Total variance explained in step 2c as a function of the number of PCs used for the reconstruction. Each thick line shows the results for one normal hearing animal (n = 6). The thin line denotes 95% variance explained. (**c**) Signal variance explained in step 3 as a function of the number of PCs used for the reconstruction. (**d**) Percent of variance explained by each PC that corresponds to neural signal (rather than neural noise) for an example animal. (**e**) Variance explained in step 4 for each pair of normal hearing animals. (**f, g**) Total variance explained in step 2c and signal variance explained in step 3 for animals with hearing loss (n = 6). (**h**) Variance explained in step 4 for each pair of animals with hearing loss and each pair of animals with different hearing status. (**i**) Distributions of variance explained in step 4 for each pair of normal hearing animals (n = 15), each pair of animals with hearing loss (n = 15), and each pair of animals with different hearing status (n = 36). Median values were compared via Kruskal–Wallis one-way ANOVA and Tukey–Kramer post hoc tests, \*\*\*p<0.001, \*\*p<0.01, \*p<0.05, ns indicates not significant. For full details of statistical tests, see *Table 1*.

The online version of this article includes the following figure supplement(s) for figure 2:

**Figure supplement 1.** Sloping mild-to-moderate sensorineural hearing loss.

demonstrating again that the signal manifold is fundamental and robust. But the similarity between the signal dynamics of normal hearing animals and animals with hearing loss was much lower than that between animals with the same hearing status (*Figure 2h and i*; 78% average variance explained). This result indicates that the activity within the signal manifold of an animal with hearing loss is not linearly predictable from the activity within the signal manifold of a normal hearing animal and, thus, that the impact of hearing loss at the network level is a true nonlinear distortion that reshapes the neural code in a complex way.

## DNNs enable accurate simulation of neural signal dynamics

To develop an understanding of exactly how hearing loss impacts signal dynamics, further investigation is required. However, traditional approaches to manifold learning such as PCA are limited by the fact that they can only be applied to existing recordings. To overcome this limitation, we designed a DNN that allowed us to identify the signal manifold within the framework of an encoding model that maps sound to neural activity (*Figure 3a*). If the DNN can be trained to replicate neural activity with high accuracy for a wide range of sounds, it can then be used to probe the effects of hearing loss on signal dynamics using new sounds as needed.

The DNN first projects sound into a high-dimensional feature space using an encoder with a cascade of convolutional layers. It then reduces the dimensionality of its feature space through a bottleneck layer and uses a simple (i.e., not convolutional) linear readout to transform the activations in the bottleneck into the neural activity for each recorded unit. During training, the DNN learns to use the bottleneck to identify the low-dimensional feature space that captures as much of the explainable variance in the recorded neural activity as possible, that is, the signal manifold. (Note that the structure of the DNN is not meant to reflect the anatomy of the auditory system; it is simply a tool for identifying latent dynamics and predicting neural activity.)

Once trained, the DNN can be used to simulate neural activity for any sound, whether it was presented during neural recordings or not, with the activations in the bottleneck layer reflecting the underlying signal dynamics. This supervised approach to identifying the signal manifold also has the added advantage that it eliminates the residual noise that is inevitable with unsupervised methods such as PCA (*Figure 3b*). (See decreasing SNR with successive PCs in *Figure 2d*.).

The utility of the DNN rests, of course, on its ability to faithfully reproduce the recorded neural activity. We trained and tested a separate DNN for each animal (after partitioning the recordings into training and test sets as described above) and found that they performed with remarkable accuracy. The explainable variance explained for activity in the test set approached 100% for the units with highest explainable variance and was far beyond that achieved by a standard single-layer linear–nonlinear model (*Figure 3c*). We varied the size of the bottleneck layer and found that performance plateaued with more than eight channels for both normal hearing animals and those with hearing loss, consistent with the dimensionality of the signal manifold identified through PCA (*Figure 3c*).

We also assessed the similarity of the DNN-derived signal dynamics across animals by measuring how much of the variance in the bottleneck activations from one animal could be explained by the bottleneck activations from another (*Figure 3d and e*; allowing for additional linear transformation).

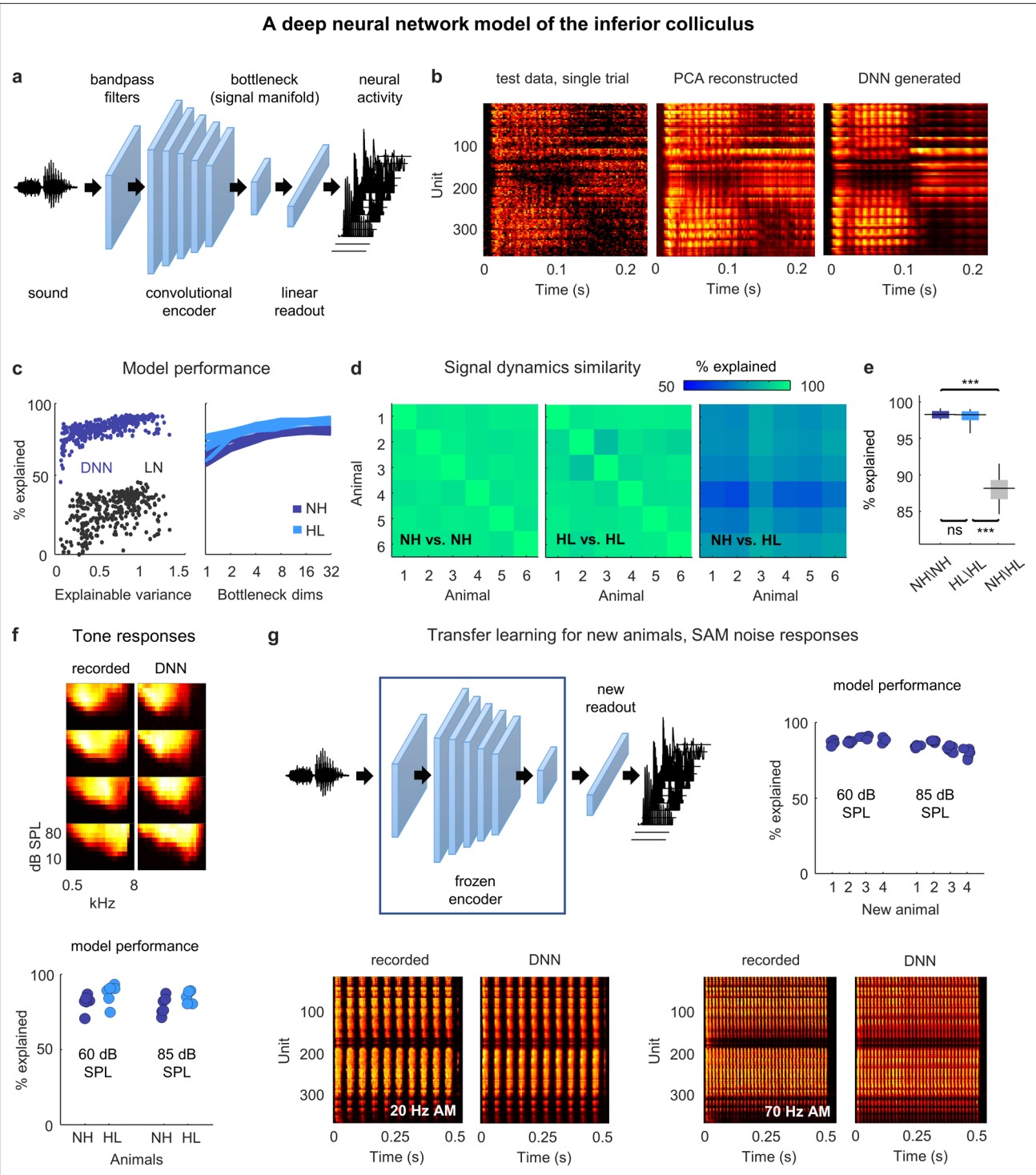

**Figure 3.** Neural signal dynamics identified via deep learning. (**a**) Schematic diagram of the deep neural network (DNN) used to predict inferior colliculus (IC) activity. (**b**) Example images of neural activity elicited by a speech syllable from an example recording (left), reconstructed as in step 2b of *Figure 2* (center), and predicted by the DNN (right). (**c**) Left: predictive power of the DNN for each unit from one animal. The value on the horizontal axis is the fraction of the variance in the neural activity that is explainable (i.e., that is consistent across trials of identical speech). The value on the vertical axis is the percent of this explainable variance that was captured by the DNN. Analogous values for linear–nonlinear (LN) models trained and tested on the same activity are shown for comparison. Right: average predictive power for all units as a function of the number of channels in the bottleneck layer. Each line shows the results for one animal. (**d**) Fraction of the variance in one set of bottleneck activations explained by another set for each pair of normal hearing animals (left), each pair of animals with hearing loss (center), and each pair of animals with different hearing status (right). (**e**) Distributions of variance in one set of bottleneck activations explained by another set for each pair of normal hearing animals (n = 15), each pair of

*Figure 3 continued on next page*

*Figure 3 continued*

animals with hearing loss (n = 15), and each pair of animals with different hearing status (n = 36). Median values were compared via Kruskal–Wallis one-way ANOVA and Tukey–Kramer post hoc tests, \*\*\*p<0.001, \*\*p<0.01, \*p<0.05, ns indicates not significant. For full details of statistical tests, see *Table 1*. (**f**) Top: example images of neural activity elicited by pure tones from an example recording (left) and predicted by the DNN (right). Each subplot shows the frequency response area for one unit (the average activity recorded during the presentation of tones with different frequencies and sound levels). The colormap for each plot is normalized to the minimum and maximum activity level across all frequencies and intensities. Bottom: predictive power of the DNN for tone responses across all frequencies at two different intensities. Each point indicates the average percent of the explainable variance that was captured by the DNN for all units from one animal (with horizontal jitter added for visibility). (**g**) Top left: schematic diagram of transfer learning for new animals. Bottom: example images of neural activity elicited by sinusoidally amplitude modulated (SAM) noise with two different modulation frequencies from an example recording (left; average over 128 repeated trials) and predicted by the DNN (right). Top right: predictive power of the DNN for SAM noise responses across all modulation frequencies and modulation depths at two different intensities. Each point indicates the average percent of the explainable variance that was captured by the DNN for all units from one of four new animals after transfer learning using a frozen encoder from one of six animals in the original training set.

We found that the signal dynamics for animals with the same hearing status were nearly identical (97% average variance explained for both normal hearing and hearing loss), while the similarity between the signal dynamics for animals with normal hearing and those with hearing loss was much lower (87% average variance explained). Thus, the signal manifold as identified by the DNN appears to have similar properties to that identified through PCA, with the added advantages of reduced residual noise and the ability to probe the signal dynamics using novel sounds.

To examine the degree to which the DNNs trained on speech were capable of predicting responses to other sounds, we compared recorded and DNN-generated responses to pure tones with different frequencies and intensities (*Figure 3f*). The DNNs performed well, explaining an average of 83% of the explainable variance in the recorded activity across animals. To further test the generality of the DNN models, we used transfer learning to test their ability to predict responses to new sounds for a new set of animals. If the DNN encoder really does capture transformations that are common to all animals with the same hearing status, then it should be possible to use a trained encoder from one animal to predict responses for a new animal after learning only a new linear readout (*Figure 3g*). For each of the DNN models trained on activity from one of the six normal hearing animals in our original dataset, we froze the encoder and retrained the linear readout for each of four new normal hearing animals. We initialized the readout weights for each unit in a new animal using the readout weights for a random unit from the original animal, and then optimized the weights using a relatively small sample (between 2 and 3.5 hr) of activity recorded from the new animal during the presentation of speech and moving ripples. We then used the new DNN model (the frozen encoder and the optimized readout) to predict responses from the new animal to sinusoidally amplitude modulated (SAM) broadband noise sounds with different modulation frequencies, modulation depths, and intensities. The new DNN models performed well, explaining an average of 85% of the explainable variance in the recorded activity across animals. While pure tones and SAM noise are only two of many possible sounds, these results provide encouraging evidence of the generality of the DNN models.

## Hearing loss distorts spectral processing

Before continuing our investigation of the neural coding of speech, we first used the DNN to examine the impact of hearing loss on the processing of basic acoustic features. To assess spectral processing, we presented the DNN for each animal with a stream of pure tones with different frequencies and intensities and extracted the activations from the bottleneck layer (*Figure 4a*; we set the dimensionality of the bottleneck layer to 8 for this and all subsequent analyses). The frequency response areas (FRAs) for individual bottleneck channels resembled those that are typically observed for individual neurons in the IC: some exhibited a clear preferred frequency at low intensities and broader tuning at high intensities, while others had more complex shapes (*Figure 4b*). For animals with hearing loss, elevated intensity thresholds were also evident.

To visualize the signal dynamics, we applied PCA to the bottleneck activations (for each intensity separately) and projected the full dynamics onto the top two PCs, which explained more than 90% of the variance for these simple sounds (*Figure 4c*). For normal hearing animals, the paths traced by the dynamics within the signal manifold for different sounds, which we term *trajectories*, were distinct and formed an orderly arrangement, but with a clear change in geometry across intensities (*Figure 4d*). At low intensities, the trajectories for different frequencies were distinct across both PCs, each of which

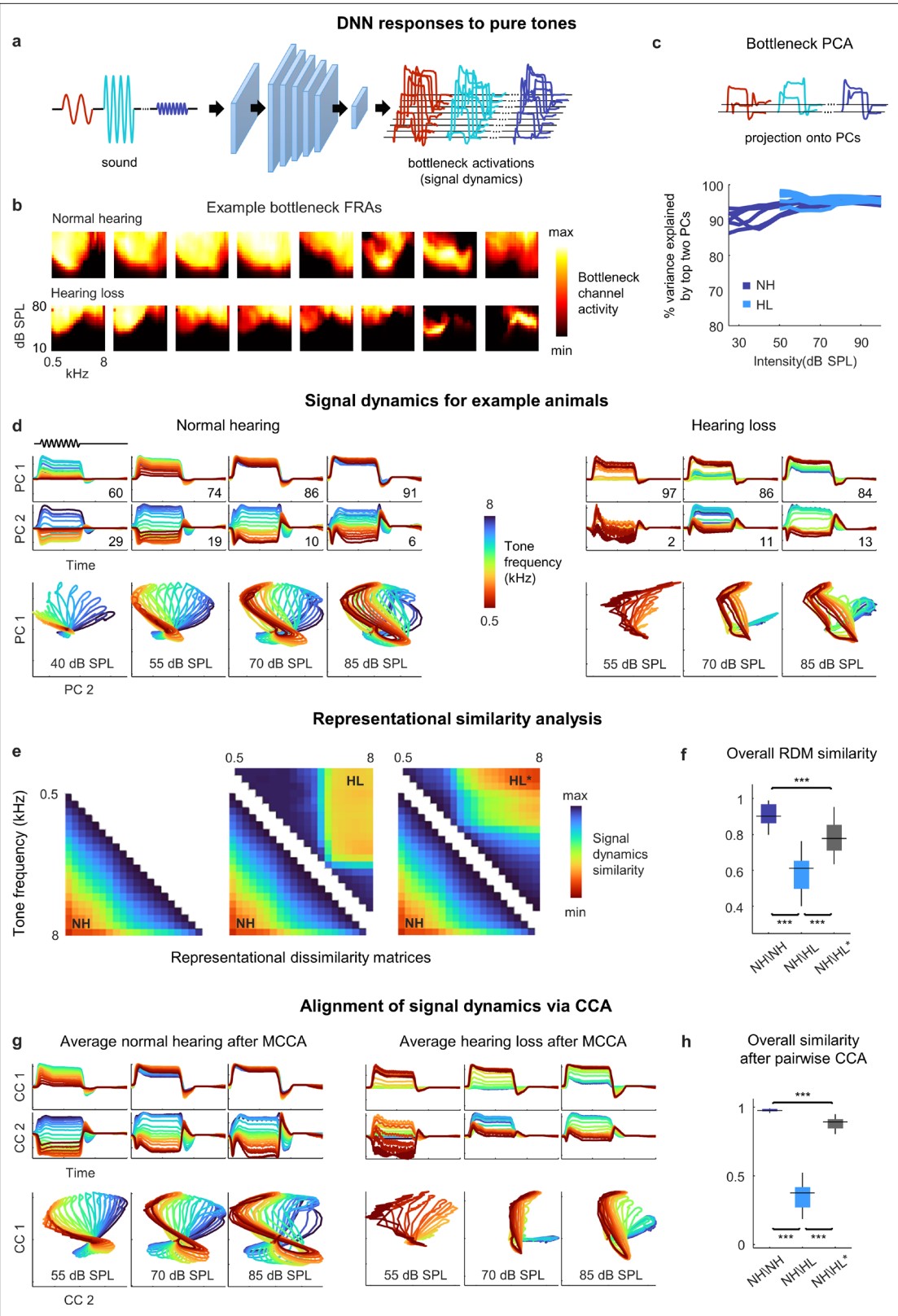

**Figure 4.** Neural signal dynamics for pure tones. (**a**) Schematic diagram showing pure tone sounds with different frequencies and intensities and corresponding bottleneck activations. (**b**) Frequency response areas (FRAs) for the eight bottleneck channels from a normal hearing animal (top) and an animal with hearing loss (bottom). Each subplot shows the average activity for one channel in response to tones with different frequencies and intensities. The colormap for each plot is normalized to the minimum and maximum activity level across all frequencies and intensities. (**c**) Dimensionality

*Figure 4 continued on next page*

*Figure 4 continued*

reduction of bottleneck activations via principal component analysis (PCA). Each line shows the variance explained by the top two principal components (PCs) for one animal as a function of the intensity of the tones. (**d**) Signal dynamics for pure tones for a normal hearing animal (left) and an animal with hearing loss (right). The top two rows show the projection of the bottleneck activations onto each of the top two PCs as a function of time. The inset value indicates the percent of the variance in the bottleneck activations explained by each PC. The bottom row shows the projections from the top two rows plotted against one another. Each line shows the dynamics for a different tone frequency. Each column shows the dynamics for a different tone intensity. (**e**) Representational dissimilarity matrices (RDMs) computed from bottleneck activations. The left image shows the average RDM for normal hearing animals for tones at 55 dB SPL. The value of each pixel is proportional to the point-by-point correlation between the activations for a pair of tones with different frequencies. The center image shows the same lower half of the RDM for normal hearing animals along with the upper half of the RDM for animals with hearing loss at the same intensity. The right image shows the same lower half of the RDM for normal hearing animals along with the upper half of the RDM for animals with hearing loss at the best intensity (that which produced the highest point-by-point correlation between the normal hearing and hearing loss RDMs). (**f**) The point-by-point correlation between RDMs for each pair of normal hearing animals (n = 15), and each pair of animals with different hearing status compared at either the same intensity or the best intensity (n = 36). Median values were compared via Kruskal–Wallis one-way ANOVA and Tukey–Kramer post hoc tests, \*\*\*p<0.001, \*\*p<0.01, \* p<0.05, ns indicates not significant. For full details of statistical tests, see *Table 1*. (**g**) Average signal dynamics for pure tones for normal hearing animals (left) and animals with hearing loss (right) after alignment via multiway canonical correlation analysis (MCCA). (**h**) The similarity between dynamics after alignment via pairwise canonical correlation analysis (CCA) (see 'Methods') for each pair of normal hearing animals, and each pair of animals with different hearing status compared at either the same intensity or the best intensity (that which produced the highest similarity between the dynamics).

The online version of this article includes the following figure supplement(s) for figure 4:

**Figure supplement 1.** Distorted spectral processing in the inferior colliculus (IC) and auditory nerve (AN).

accounted for substantial variance (the percent of the variance in the signal dynamics explained by each PC at each intensity is indicated on each panel). But at high intensities, the trajectories for all frequencies were similar along the first PC, which accounted for the vast majority of the variance, and varied only along the second PC. These intensity-dependent changes in the geometry of the signal dynamics are consistent with the known effects of intensity on spectral tuning in IC neurons. At low intensities, tuning is narrow and, thus, different tone frequencies elicit distinct population activity patterns. But at high intensities, because tuning is broader, the population activity patterns elicited by different frequencies are less distinct.

For animals with hearing loss, the signal dynamics were dramatically different. At moderate intensity, only the lowest frequencies elicited any activation (as expected, given the larger elevation in intensity thresholds at high frequencies; *Figure 2—figure supplement 1*). At higher intensity, all tones elicited activation, but rather than forming an orderly arrangement within the manifold, the trajectories for different frequencies clustered into two groups, one for low frequencies and one for high frequencies. At the highest intensity, the trajectories for different frequencies became more distinct, but clear clustering remained. The increased similarity in the signal trajectories for different frequencies is consistent with the increased similarity of spectral tuning between individual IC neurons in animals with hearing loss (*Barsz et al., 2007*; *Willott, 1986*), but the clustering of dynamics within the signal manifold is an emergent network-level phenomenon that has not previously been observed.

To further analyze the dynamics and allow for direct comparisons across animals, we first turned to representational similarity analysis (RSA) (*Kriegeskorte et al., 2008*). RSA uses only the relative distances between trajectories within a manifold and, thus, does not require the dynamics for different animals to be aligned. The first step in RSA is to form a representational dissimilarity matrix (RDM) for each set of trajectories, with each entry in the RDM equal to one minus the point-by-point correlation between a pair of trajectories for two different sounds (*Figure 4e*, left).

The structure of the RDMs was consistent with the observations about the dynamics made above. For normal hearing animals, the RDMs had a diagonal symmetry with a smooth gradient, indicating that the similarity of the trajectories for two frequencies decreased gradually as the difference between the frequencies increased. For animals with hearing loss, the RDMs had a block structure, indicating that the trajectories formed two clusters (*Figure 4e*, center; note that RDMs are symmetric, so the lower half of the normal hearing RDM is shown with the upper half of the hearing loss RDM for comparison).

Because we were interested in identifying the effects of hearing loss beyond those related to audibility, we also compared the normal hearing RDMs at a given intensity to the hearing loss RDMs at the *best intensity*, that is, at whatever intensity resulted in the highest similarity to the normal hearing RDM for each pair of animals (measured as the point-by-point correlation between the RDMs). Amplification to the best intensity softened the block structure and shifted the transition between clusters

to a lower frequency, but did not fully restore the diagonal structure present in the normal hearing RDMs (*Figure 4e*, right). Overall, the similarity between the RDMs for different normal hearing animals at moderate intensity (55 dB SPL) was high (0.91 ± 0.01; mean ± SEM; n = 15 pairwise comparisons; *Figure 4f*; for full details of all statistical tests, see *Table 1*). The similarity between the normal hearing and hearing loss RDMs at the same moderate intensity was much lower (0.59 ± 0.02; n = 36) and remained relatively low even at the best intensity (0.78 ± 0.02; n = 36).

The results of RSA are easy to interpret, but, because it uses only the relative distances between trajectories, it can be insensitive to distortions that impact the overall structure of the dynamics (e.g., a change in temporal dynamics that is common across all sound frequencies). To allow for direct comparisons of overall structure, we used canonical correlation analysis (CCA) (*Dabagia et al., 2022*). CCA identifies a series of linear projections, known as canonical components (CCs), that attempt to align two sets of dynamics such that the point-by-point correlation between trajectories after projection onto their respective CCs is maximized. The set of CCs for a given set of dynamics are required to be orthogonal to each other and to explain all of the variance in the original trajectories. CCA can also be extended to simultaneously align dynamics for an entire group of animals through multiway canonical correlation analysis (MCCA) (*de Cheveigné et al., 2019*).

The average dynamics after alignment via CCA exhibited phenomena that were similar to those that were evident for individual animals (*Figure 4g*). For normal hearing animals, the trajectories for different tone frequencies were distinct and formed an orderly arrangement with frequency-dependent variation across two dimensions at low intensities, while at high intensities the variation across frequencies was largely confined to the second CC. For animals with hearing loss, the trajectories for different frequencies clustered into two groups at moderate intensities and remained clustered, albeit more weakly, at high intensities. This clustering was also evident when a similar analysis was performed directly on recorded neural activity (*Figure 4—figure supplement 1*).

To measure the similarity between the dynamics for different animals after alignment via CCA, we used a weighted sum of the point-by-point correlations between the two sets of dynamics after projection onto each pair of CCs, with the weight for the correlation associated with each pair of CCs given by the average variance in the original dynamics that those CCs explained (see 'Methods' for equation). Overall, the similarity between the dynamics for different normal hearing animals at moderate intensity after alignment via CCA was extremely high (0.98 ± 0.01; n = 15; *Figure 4h*). The similarity between the aligned dynamics for normal hearing and hearing loss animals at the same moderate intensity was much lower (0.37 ± 0.02; n = 36) and remained below normal even when compared at the best intensity (0.88 ± 0.01; n = 36). Taken together, the RSA and CCA results suggest that hearing loss results in a fundamental disruption of spectral processing at the network level.

## Hearing loss does not distort temporal processing

We next assessed temporal processing by performing a similar analysis on the bottleneck activations elicited by a stream of SAM broadband noise sounds with different modulation frequencies and intensities (*Figure 5a*). For these sounds, two dimensions were again enough to capture almost all of the variance in the full signal dynamics across all intensities (*Figure 5b*). For both normal hearing animals and those with hearing loss, the explicit tracking of envelope modulations in the signal dynamics decreased with increasing modulation frequency and increasing intensity (*Figure 5c*). But when compared at the same intensity, the dynamics for animals with hearing loss clearly differed from those for animals with normal hearing (*Figure 5d and e*).

This was confirmed by RSA (*Figure 5f*), which indicated that while the similarity between normal hearing RDMs at moderate intensity (55 dB SPL) was extremely high (0.99 ± 0.01; n = 15), the similarity between normal hearing and hearing loss RDMs was lower (0.76 ± 0.01; n = 36). But when compared at the best intensity to eliminate differences related to audibility, the dynamics for animals with hearing loss became nearly identical to those for animals with normal hearing (*Figure 5e*), and this was reflected in the similarity between RDMs (0.99 ± 0.01; n = 36).

Comparing the similarity of the dynamics after alignment via CCA yielded similar results (*Figure 5g*). The similarity between the dynamics for different normal hearing animals at moderate intensity after alignment via CCA was high (0.97 ± 0.01; n = 15). The similarity between the aligned dynamics for normal hearing and hearing loss animals was much lower when compared at the same moderate intensity (0.44 ± 0.02; n = 36) but increased to normal levels when the comparison was made at the

**Table 1.** Details of statistical analyses.

This table provides the details for the statistical analyses in this study, including sampling unit, sample sizes, and p-values. All comparisons were made using Kruskal–Wallis one-way ANOVA with post hoc Tukey–Kramer tests to compute pairwise p-values.

### Figure 2

**Figure 2i** — Sampling unit: pairs of animals

| Groups: | Comparisons: | |
|---|---|---|
| 1. NH\NH (n = 15) | 1 vs. 2 | p=0.08 |
| 2. HL\HL (n = 15) | 1 vs. 3 | p<1e-7 |
| 3. NH\HL (n = 36) | 2 vs. 3 | p<1e-7 |

### Figure 3

**Figure 3e** — Sampling unit: pairs of animals

| Groups: | Comparisons: | |
|---|---|---|
| 1. NH\NH (n = 15) | 1 vs. 2 | p=0.71 |
| 2. HL\HL (n = 15) | 1 vs. 3 | p<1e-10 |
| 3. NH\HL (n = 36) | 2 vs. 3 | p<1e-10 |

### Figure 4

**Figure 4f** — Sampling unit: pairs of animals

| Groups: | Comparisons: | |
|---|---|---|
| 1. NH\NH (n = 15) | 1 vs. 2 | p<1e-7 |
| 2. NH\HL (n = 36) | 1 vs. 3 | p<1e-3 |
| 3. NH\HL* (n = 36) | 2 vs. 3 | p<1e-7 |

**Figure 4f** — Sampling unit: pairs of animals

| Groups: | Comparisons: | |
|---|---|---|
| 1. NH\NH (n = 15) | 1 vs. 2 | p<1e-7 |
| 2. NH\HL (n = 36) | 1 vs. 3 | p<1e-4 |
| 3. NH\HL* (n = 36) | 2 vs. 3 | p<1e-7 |

### Figure 5

**Figure 5f** — Sampling unit: pairs of animals

| Groups: | Comparisons: | |
|---|---|---|
| 1. NH\NH (n = 15) | 1 vs. 2 | p<1e-7 |

### Figure 6

**Figure 6e** — Sampling unit: pairs of animals

| Groups: | Comparisons: | |
|---|---|---|
| 1. NH\NH (n = 15) | 1 vs. 2 | p<1e-7 |
| 2. NH\HL (n = 36) | 1 vs. 3 | p=0.007 |
| 3. NH\HL* (n = 36) | 1 vs. 4 | p=0.23 |
| 4. NH\HA (n = 36) | 2 vs. 3 | p<1e-7 |
| | 2 vs. 4 | p<1e-7 |
| | 3 vs. 4 | p=0.29 |

**Figure 6f** — Sampling unit: pairs of animals

| Groups: | Comparisons: | |
|---|---|---|
| 1. NH\NH (n = 15) | 1 vs. 2 | p<1e-7 |
| 2. NH\HL (n = 36) | 1 vs. 3 | p<1e-7 |
| 3. NH\HL* (n = 36) | 1 vs. 4 | p=0.002 |
| 4. NH\HA (n = 36) | 2 vs. 3 | p<1e-7 |
| | 2 vs. 4 | p<1e-7 |
| | 3 vs. 4 | p<1e-4 |

**Figure 6g** — Sampling unit: animals

| Groups: | Comparisons: | |
|---|---|---|
| 1. NH (n = 6) | 1 vs. 2 | p<1e-6 |
| 2. HL (n = 6) | 1 vs. 3 | p=0.011 |
| 3. HL* (n = 6) | 1 vs. 4 | p=0.057 |
| 4. HA (n = 6) | 2 vs. 3 | p<1e-3 |
| | 2 vs. 4 | p<1e-4 |
| | 3 vs. 4 | p=0.86 |

### Figure 7

**Figure 7e** — Sampling unit: pairs of animals

| Groups: | Comparisons: | |
|---|---|---|
| 1. NH\NH (n = 15) | 1 vs. 2 | p<1e-7 |
| 2. NH\HL (n = 36) | 1 vs. 3 | p<1e-4 |
| 3. NH\HL* (n = 36) | 1 vs. 4 | p<1e-7 |

*Table 1 continued on next page*

*Table 1 continued*

**Figure 2**

| Figure 2i | Sampling unit: pairs of animals | | | Figure 6 | | | |
| --- | --- | --- | --- | --- | --- | --- | --- |
| | | | | Figure 6e | Sampling unit: pairs of animals | | |
| Groups: | | Comparisons: | | Groups: | | Comparisons: | |
| 2. NH\HL (n = 36) | | 1 vs. 3 | p=0.99 | 4. NH\HA (n = 36) | | 2 vs. 3 | p<1e-7 |
| 3. NH\HL* (n = 36) | | 2 vs. 3 | p<1e-7 | | | 2 vs. 4 | p<1e-7 |
| | | | | | | 3 vs. 4 | p=0.056 |

| *Figure 5g* | Sampling unit: pairs of animals | | | | | | |
| --- | --- | --- | --- | --- | --- | --- | --- |
| | | | | *Figure 7f* | Sampling unit: pairs of animals | | |
| Groups: | | Comparisons: | | | | | |
| 1. NH\NH (n = 15) | | 1 vs. 2 | p<1e-7 | Groups: | | Comparisons: | |
| 2. NH\HL (n = 36) | | 1 vs. 3 | p=0.64 | 1. NH\NH (n = 15) | | 1 vs. 2 | p<1e-7 |
| 3. NH\HL* (n = 36) | | 2 vs. 3 | p<1e-7 | 2. NH\HL (n = 36) | | 1 vs. 3 | p<1e-7 |
| | | | | 3. NH\HL* (n = 36) | | 1 vs. 4 | p<1e-7 |
| *Figure 5j* | Sampling unit: pairs of animals | | | 4. NH\HA (n = 36) | | 2 vs. 3 | p<1e-7 |
| | | | | | | 2 vs. 4 | p<1e-7 |
| Groups: | | Comparisons: | | | | 3 vs. 4 | p=0.99 |
| 1. NH\NH (n = 15) | | 1 vs. 2 | p<1e-7 | | | | |
| 2. NH\HL (n = 36) | | 1 vs. 3 | p=0.89 | *Figure 7g* | Sampling unit: animals | | |
| 3. NH\HL* (n = 36) | | 2 vs. 3 | p<1e-7 | | | | |
| | | | | Groups: | | Comparisons: | |
| *Figure 5k* | Sampling unit: pairs of animals | | | 1. NH (n = 6) | | 1 vs. 2 | p<1e-9 |
| | | | | 2. HL (n = 6) | | 1 vs. 3 | p<1e-5 |
| Groups: | | Comparisons: | | 3. HL* (n = 6) | | 1 vs. 4 | p<1e-6 |
| 1. NH\NH (n = 15) | | 1 vs. 2 | p<1e-7 | 4. HA (n = 6) | | 2 vs. 3 | p<1e-4 |
| 2. NH\HL (n = 36) | | 1 vs. 3 | p=0.97 | | | 2 vs. 4 | p<1e-3 |
| 3. NH\HL* (n = 36) | | 2 vs. 3 | p<1e-7 | | | 3 vs. 4 | p=0.59 |

NH: normal hearing; HL: hearing loss; HL*: hearing loss at best intensity; HA: hearing aid.

best intensity (0.95 ± 0.01; n = 36). Thus, it appears that hearing loss has little impact on temporal processing beyond that which results from decreased audibility.

We verified that this was also true for the processing of sounds with different modulation depths. We performed the same analysis on the bottleneck activations elicited by a stream of SAM noise sounds with different modulation depths and intensities (and a fixed modulation frequency of 30 Hz; *Figure 5h*). When compared at the best intensity, the signal dynamics for normal hearing animals and animals with hearing loss were nearly identical (*Figure 5i*), with the explicit tracking of envelope modulations decreasing with decreasing modulation depth. The overall similarity measured at the best intensity both by RSA (0.99 ± 0.01; n = 36; *Figure 5j*) and after alignment via CCA (0.96 ± 0.01; n = 36; *Figure 5k*) confirmed that the impact of hearing loss on temporal processing beyond that which results from decreased audibility was negligible.

## Distortions in the neural code for speech in quiet are largely corrected by amplification

Having established that the distortions in neural signal dynamics caused by hearing loss affect primarily spectral, rather than temporal, processing for simple sounds, we next returned to speech. We focused

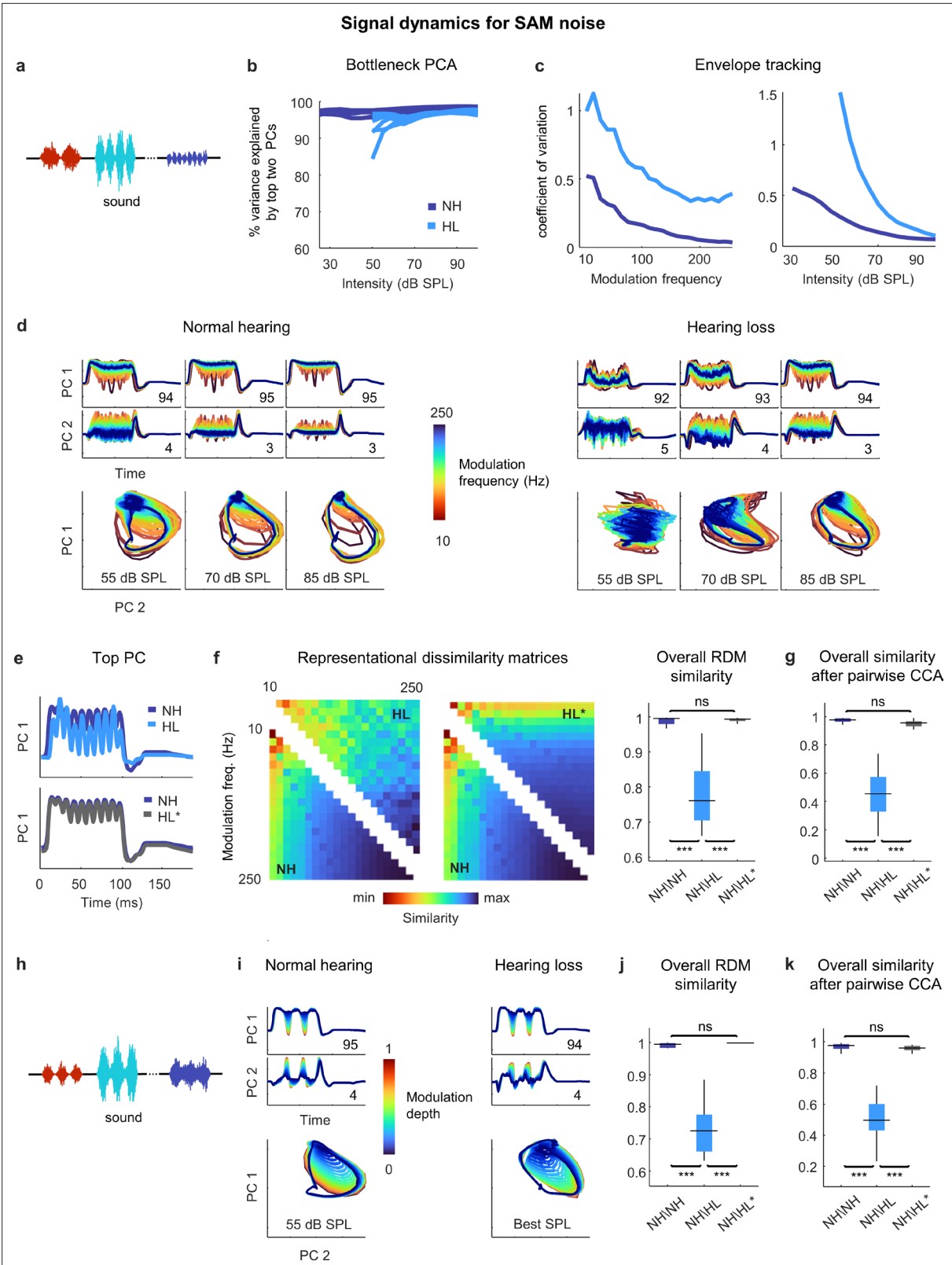

**Figure 5.** Neural signal dynamics for amplitude modulated noise. (**a**) Schematic diagram showing amplitude-modulated noise sounds with different intensities and modulation frequencies. (**b**) Variance in bottleneck activations explained by top two principal components (PCs) for each animal as function of sound intensity. (**c**) Envelope tracking in signal dynamics measured as the coefficient of variation of the bottleneck activations for sounds with different modulation frequencies at an intensity of 55 dB SPL (left) and sounds at different intensities with a modulation frequency of 100 Hz

*Figure 5 continued*

(right). Values shown are the average across animals. (**d**) Signal dynamics for a normal hearing animal (left) and an animal with hearing loss (right). Each line shows the dynamics for a different modulation frequency. (**e**) Signal dynamics for a modulation frequency of 100 Hz after projection onto the top PC. The top panel shows the dynamics for a normal hearing animal and an animal with hearing loss at 55 dB SPL. The bottom panel shows the same dynamics for the normal hearing animal along with the dynamics for the animal with hearing loss at the best intensity. (**f**) Representational dissimilarity matrices (RDMs) computed from bottleneck activations and the point-by-point correlation between RDMs for different pairs of animals at 55 dB SPL or best intensity. Median values were compared via Kruskal–Wallis one-way ANOVA and Tukey–Kramer post hoc tests, \*\*\*p<0.001, \*\*p<0.01, \*p<0.05, ns indicates not significant. For full details of statistical tests, see *Table 1*. (**g**) The similarity between dynamics after alignment via pairwise canonical correlation analysis (CCA) for different pairs of animals at 55 dB SPL or best intensity. (**h**) Schematic diagram showing amplitude modulated noise sounds with different intensities and modulation depths. (**i**) Signal dynamics for a normal hearing animal at 55 dB SPL (left) and an animal with hearing loss at the best intensity (right). Each line shows the dynamics for a different modulation depth. (**j**) The point-by-point correlation between RDMs for different pairs of animals at 55 dB SPL or best intensity. (**k**) The similarity between dynamics after alignment via pairwise CCA for different pairs of animals at 55 dB SPL or best intensity.

on consonants, which vary widely in their spectral properties and are the primary contributor to the perceptual deficits exhibited by people with hearing loss when listening to ongoing speech (*Fogerty et al., 2012*). We presented the DNN with a stream of isolated consonants (diphone syllables with the vowel removed), each uttered multiple times by multiple talkers (*Figure 6a*). The consonants can be divided into three broad groups: the vowel-like consonants (nasals and approximants), which are dominated by low frequencies; the plosives, which are broadband; and the fricatives, which are dominated by high frequencies (*Figure 6b*).

For both normal hearing animals and animals with hearing loss, two dimensions were again sufficient to explain nearly all of the variance in the bottleneck activations (*Figure 6c*). For normal hearing animals, the dynamics elicited by different consonants followed distinct trajectories that were organized within the signal manifold according to consonant type (*Figure 6d*; the dynamics shown are the average across all instances of each consonant). For animals with hearing loss, only the vowel-like consonants elicited responses at moderate intensity (as expected, given the larger elevation in intensity thresholds at high frequencies; *Figure 2—figure supplement 1*). At higher intensities, all consonants elicited responses but the trajectories were not as distinct as with normal hearing and exhibited a clustering similar to that observed for pure tones (*Figure 4d*), which softened at the highest intensity.

The differences in the dynamics for normal hearing animals and those with hearing loss were evident in the RDMs (*Figure 6e*). When compared at a typical conversational intensity (60 dB SPL), the similarity between normal hearing RDMs was high (0.94 ± 0.01; n = 15), but the similarity between normal hearing and hearing loss RDMs was low (0.23 ± 0.02; n = 36). The normal hearing and hearing loss RDMs were much more similar when compared at the best intensity, though some differences remained (0.87 ± 0.01; n = 36).

Comparing the similarity of the dynamics after alignment via CCA yielded similar results (*Figure 6f*). The similarity between the dynamics for different normal hearing animals after alignment via CCA was high (0.96 ± 0.01; n = 15). The similarity between normal hearing and hearing loss animals when compared at the same conversational intensity was much lower (0.31 ± 0.02; n = 36) and increased when the comparison was made at the best intensity, but not to normal levels (0.77 ± 0.01; n = 36).

Given that hearing loss seems to impact primarily spectral processing, we investigated whether the distortions in the neural code for speech could be corrected by providing frequency-dependent amplification using a simulated hearing aid (*Armstrong et al., 2022*; *Alexander and Masterson, 2015*). We used the measured ABR threshold shifts for each animal to set the parameters for the amplification, which resulted in a gain of approximately 10 dB at low frequencies and 30 dB at high frequencies for speech at conversational intensity (*Figure 2—figure supplement 1*), and presented the same stream of consonants again after processing with the hearing aid. The frequency-dependent amplification was effective in reducing the distortion in the dynamics for animals with hearing loss. The overall similarity between normal hearing and hearing loss animals as measured by RSA was restored to normal levels (0.91 ± 0.01; n = 36; *Figure 6e*), and the similarity measured after alignment via CCA was also increased, though some residual distortion remained (0.86 ± 0.01; n = 36; *Figure 6f*). (Note that we did not record neural activity in response to speech through the simulated hearing aid; while we have no specific reason to doubt the accuracy of the DNN model for this class of sounds, the fact that it has not been explicitly validated should be considered.)

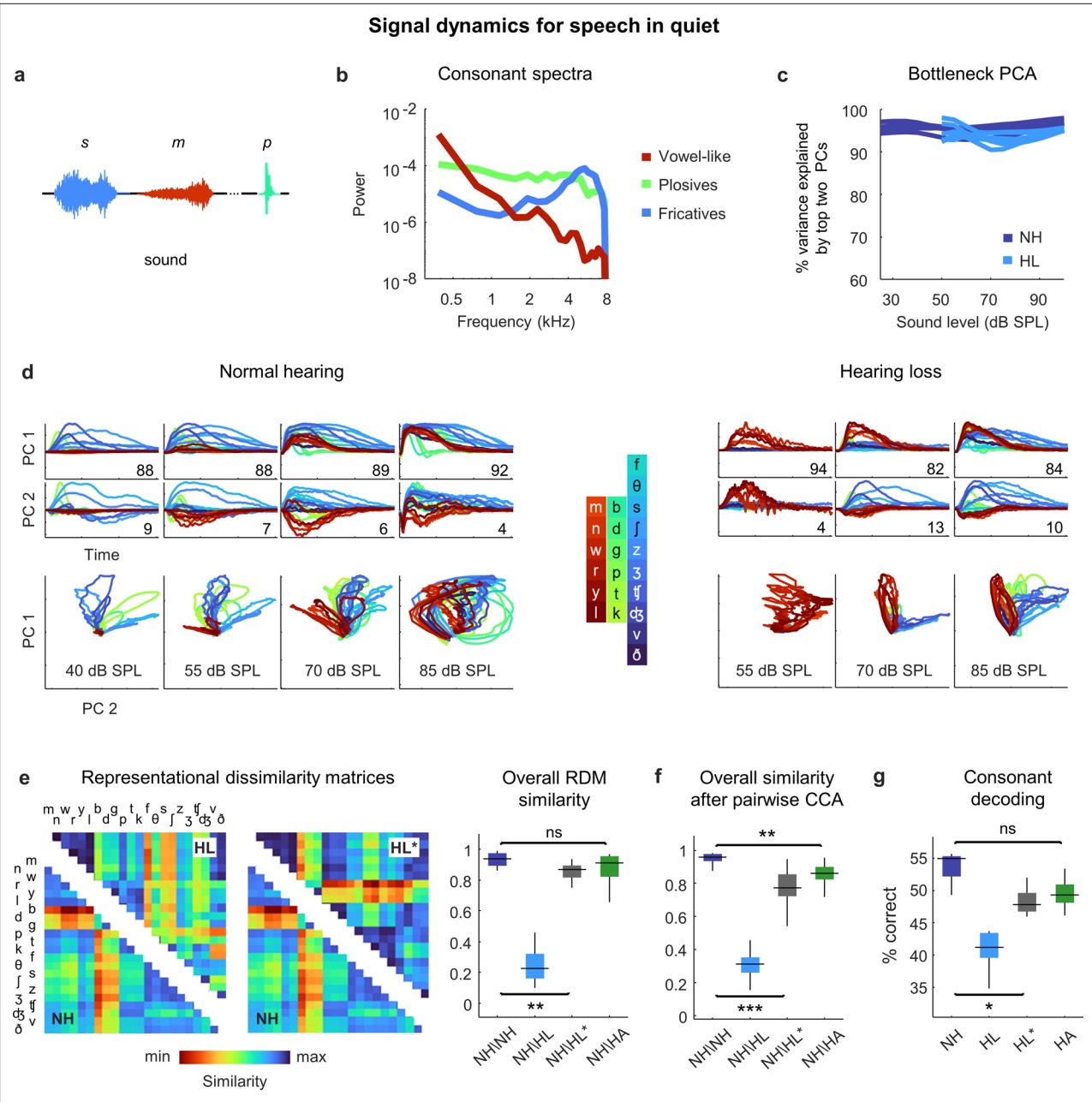

**Figure 6.** Neural signal dynamics for speech in quiet. (**a**) Schematic diagram showing different consonants. (**b**) Average power spectra for three classes of consonants. The individual consonants are shown in the inset in panel (**d**). (**c**) Variance in bottleneck activations explained by top two principal components (PCs) for each animal as a function of sound intensity. (**d**) Signal dynamics for a normal hearing animal (left) and an animal with hearing loss (right). Each line shows the dynamics for a different consonant, averaged over all instances. (**e**) Representational dissimilarity matrices (RDMs) computed from bottleneck activations and the point-by-point correlation between RDMs for different pairs of animals at 60 dB SPL, best intensity, or with a hearing aid. Median values were compared via Kruskal–Wallis one-way ANOVA and Tukey–Kramer post hoc tests, ***p<0.001, **p<0.01, *p<0.05, ns indicates not significant. For full details of statistical tests, see *Table 1*. (**f**) The similarity between dynamics after alignment via pairwise canonical correlation analysis (CCA) for different pairs of animals at 60 dB SPL, best intensity, or with a hearing aid. (**g**) Performance of a support vector machine classifier trained to identify consonants based on bottleneck activations for each normal hearing animal (n = 6) at 60 dB SPL and each animal with hearing loss (n = 6) at either 60 dB SPL, best intensity, or with a hearing aid.

To evaluate the functional consequences of the remaining distortion, we turned to decoding. We trained a support vector machine to classify consonants based on the signal dynamics for each animal. For normal hearing animals, the decoder identified 55% of consonants correctly (±1%; n = 6; chance = 4.5%) when the consonants were presented at conversational intensity (*Figure 6g*). For animals with

hearing loss, performance at the same intensity was lower (41 ± 1%; n = 6) but increased substantially at best intensity (47 ± 1%; n = 6), and increased further still with frequency-dependent amplification by the hearing aid (49 ± 1%; n = 6). Taken together, these results suggest that while amplification cannot completely restore the neural code for speech in quiet to normal, the residual distortions are relatively minor.

## Distortions in the neural code for speech in noise persist even after frequency-weighted amplification

Given that the perceptual difficulties experienced by listeners with hearing loss are most pronounced in noisy environments, we expected that the addition of background noise to the speech would create larger distortions in the neural code. We presented the same consonant stream with added speech babble (background noise formed by adding together the voices of many different talkers; *Figure 7a and b*) at a speech-to-noise ratio of 3 dB, which is typical of real-world settings experienced by hearing aid users (*Christensen et al., 2021*). The addition of background noise increased the dimensionality of the signal dynamics relative to simple sounds or speech in quiet, especially at high overall intensities; three PCs were often required to capture more than 90% of the variance (*Figure 7c*). (Note that both the speech and the background noise contribute to the signal dynamics, which encode all incoming sounds without distinction.)

For normal hearing animals, the signal dynamics for speech-in-noise and speech-in-quiet were nearly identical at the lowest intensities, but differed strongly at higher intensities (*Figure 7d*; the dynamics shown are the average across all instances of each consonant). For speech-in-noise at the highest intensity, there was a clear distinction between the first PC, which provided a clean reflection of each consonant (though not the same as for speech in quiet), and the other PCs, which were dominated by the background noise (despite averaging across all instances of each consonant with independent noise). The trends were similar for animals with hearing loss, though the background noise was reflected in the signal dynamics even more strongly.

When compared at the same high intensity (70 dB SPL), typical of a social setting, both RSA and CCA indicated strong effects of hearing loss on the signal dynamics for speech in noise (*Figure 7e and f*). The similarity between normal hearing RDMs was high (0.89 ± 0.02; n = 15), but the similarity between normal hearing and hearing loss RDMs was much lower (0.15 ± 0.03; n = 36). Amplification to best intensity increased the similarity between normal hearing and hearing loss RDMs (0.62 ± 0.02; n = 36), as did the frequency-weighted amplification provided by the hearing aid (0.56 ± 0.03; n = 36), but neither was sufficient to bring the similarity close to normal levels. For both forms of amplification, the similarity of the signal dynamics for speech in noise to those with normal hearing was much lower than for speech in quiet (best intensity: 0.62 vs. 0.87, n = 36; p<1e-10, paired *t*-test; hearing aid: 0.56 vs. 0.91, n = 36, p<1e-10, paired *t*-test). Comparing the similarity of the dynamics after alignment via CCA yielded similar results. The similarity between the dynamics for different normal hearing animals after alignment via CCA was high (0.92 ± 0.01; n = 15). The similarity between normal hearing and hearing loss animals when compared at the same intensity was much lower (0.49 ± 0.02; n = 36) and increased when the comparison was made at the best intensity (0.68 ± 0.01; n = 36) or after processing with the hearing aid (0.70 ± 0.02; n = 36), but remained well below normal levels. Again, for both forms of amplification, the similarity of the signal dynamics for speech in noise to those with normal hearing was much lower than for speech in quiet (best intensity: 0.68 vs. 0.77, n = 36; p<1e-6, paired *t*-test; hearing aid: 0.70 vs. 0.86, n = 36, p<1e-10, paired *t*-test).

Decoding the signal dynamics for each animal suggested that the distortions in the signal dynamics for speech in noise had functional consequences (*Figure 7g*). For normal hearing animals, the decoder identified 32% of consonants correctly (±1%; n = 6). For animals with hearing loss, performance at the same intensity was lower (15 ± 1%; n = 6) and remained well below normal levels both at the best intensity (23 ± 1%; n = 6) or after processing with the hearing aid (22 ± 1%; n = 6).

## Hearing loss causes hypersensitivity to background noise

To gain a better understanding of the differential impact of the background noise with and without hearing loss, we used MCCA to jointly align the signal dynamics for all animals with normal hearing and hearing loss so that we could make direct comparisons. We first analyzed the results for speech in quiet. When compared at best intensity (*Figure 8a*), there was good alignment between the dynamics

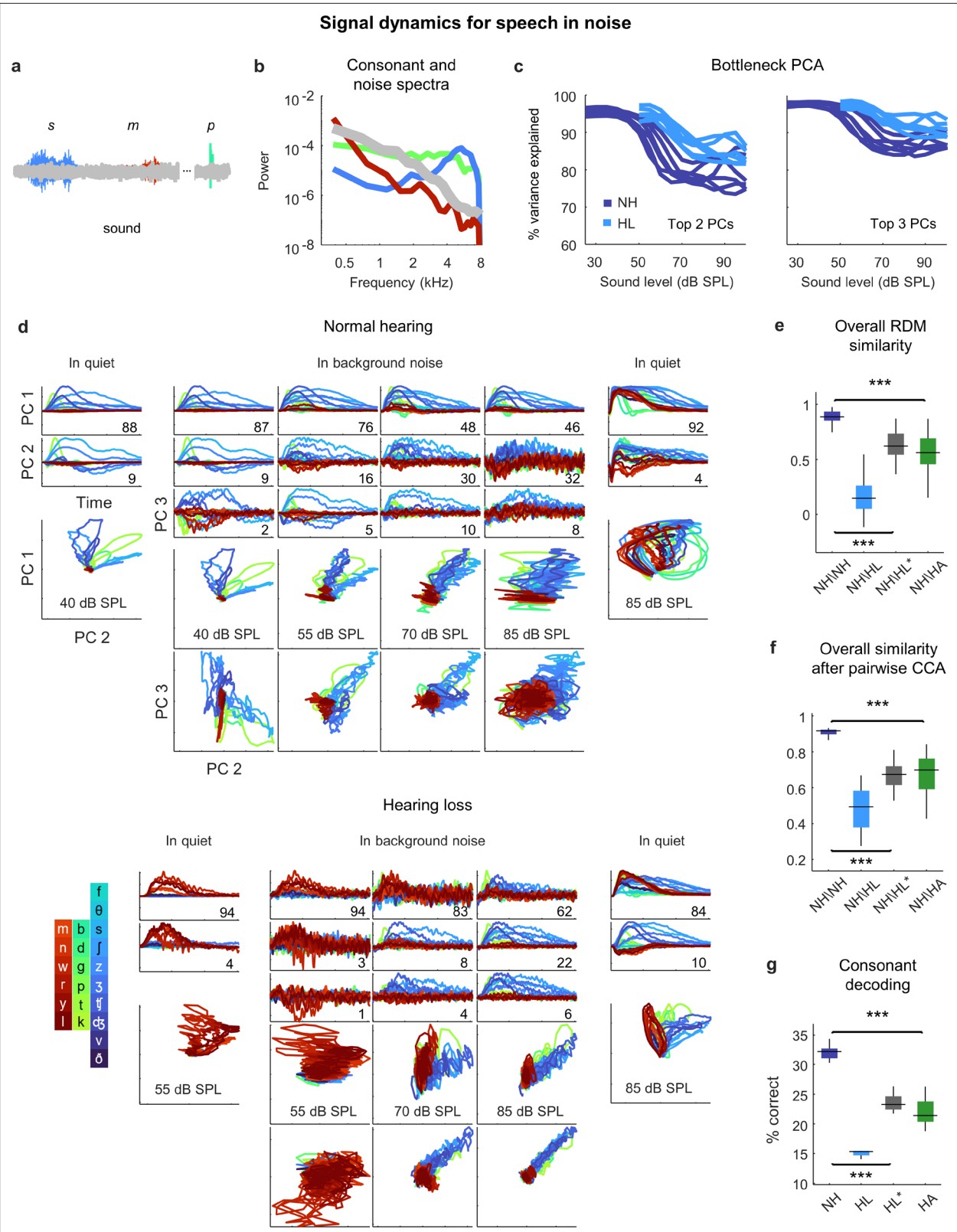

**Figure 7.** Neural signal dynamics for speech in noise. (**a**) Schematic diagram showing different consonants in speech babble. (**b**) Average power spectra for three classes of consonants and speech babble. (**c**) Variance in bottleneck activations explained by top two (left) or three (right) principal components (PCs) for each animal as a function of sound intensity. (**d**) Signal dynamics for a normal hearing animal (top) and an animal with hearing loss (bottom). Each line shows the dynamics for a different consonant, averaged over all instances. Dynamics for speech in quiet are shown alongside those for speech

*Figure 7 continued on next page*

*Figure 7 continued*

in noise for comparison. (**e**) Point-by-point correlation between representational dissimilarity matrices (RDMs) for different pairs of animals at 70 dB SPL, best intensity, or with a hearing aid. Median values were compared via Kruskal–Wallis one-way ANOVA and Tukey–Kramer post hoc tests, \*\*\*p<0.001, \*\* p<0.01, \*p<0.05, ns indicates not significant. For full details of statistical tests, see *Table 1*. (**f**) The similarity between dynamics after alignment via pairwise canonical correlation analysis (CCA) for different pairs of animals at 70 dB SPL, best intensity, or with a hearing aid. (**g**) Performance of a support vector machine classifier trained to identify consonants based on bottleneck activations for each normal hearing animal at 70 dB SPL, and each animal with hearing loss at 70 dB SPL, best intensity, or with a hearing aid.

for animals with normal hearing and hearing loss. The correlation for pairs of animals after projection onto the first CC, which accounted for 88 ± 2% and 71 ± 7% of the variance in animals with normal hearing (n = 6) and hearing loss (n = 6), respectively, was 0.94 ± 0.01 (n = 36). The correlation after projection onto the second CC, which accounted for the remaining variance, was lower (0.44 ± 0.03; n = 36).

With the hearing aid (*Figure 8b*), the alignment between the dynamics for animals with normal hearing and hearing loss was even better. The correlation after projection onto the first CC remained high (0.94 ± 0.01; n = 36) and the correlation after projection onto the second CC rose substantially (0.73 ± 0.01; n = 36). These results are consistent with the analysis for speech in quiet above that suggested that only minimal distortions in the signal dynamics for animals with hearing loss remain after frequency-weighted amplification is provided (*Figure 6*).

For speech in noise, the alignment was generally worse overall. When compared at best intensity (*Figure 8c*), the correlations after projection onto the CCs were 79 ± 1%, 74 ± 3%, and 18 ± 3%, respectively (n = 36). But even for the first two CCs for which the alignment was relatively good, the percent of the variance accounted for by each was substantially different for normal hearing animals and those with hearing loss. The first CC, which provided a clean reflection of each consonant, accounted for 56 ± 1% of the variance for normal hearing animals (n = 6), but only 38 ± 7% of the variance for animals with hearing loss (n = 6). Conversely, the second CC, which was dominated by the background noise, accounted for 41 ± 6% of the variance for animals with hearing loss (n = 6), but only 28 ± 1% of the variance for animals with normal hearing. Thus, while the neural subspaces encoding the speech and the background noise seem to be similar for all animals, the balance of variance between these subspaces is tilted much more toward background noise in animals with hearing loss.

Given the nature of the hearing loss (larger elevation in intensity thresholds at high frequencies; *Figure 2—figure supplement 1*) and the spectral properties of speech babble (higher power at low frequencies; *Figure 7b*), the hypersensitivity of animals with hearing loss to the background noise is somewhat expected. However, the problem was, if anything, exacerbated by the selective amplification of high frequencies provided by the hearing aid (*Figure 8d*). The CC that was dominated by the background noise (now the first CC, since it produced the highest correlation) accounted for 46 ± 6% of the variance for animals with hearing loss (n = 6), but only 31 ± 1% of the variance for animals with normal hearing, while the CC that provided a clean reflection of each consonant (now the second CC) accounted for 56 ± 1% of the variance for normal hearing animals (n = 6), but only 33 ± 6% of the variance for animals with hearing loss (n = 6). Thus, it appears that hearing loss allows the signal dynamics to be captured by background noise at the expense of foreground speech in a way that cannot be corrected by simple frequency-dependent amplification.

To characterize the distortions in spectral processing that underlie this effect, we examined how the processing of one narrowband sound is impacted by the presence of another. We used narrowband noise modulated by a 20 Hz sinusoidal envelope as the target sound and narrowband noise modulated by a pink noise envelope as the masker (*Figure 8e*). We varied the center frequency of the target and masker separately and the intensity of the target and masker together (to maintain a constant target-to-masker ratio). We presented the target sounds to the DNN for each animal with and without the masker sounds and measured the differences between the signal dynamics across the two conditions by computing their correlation.

For animals with normal hearing, the correlation between the target only and target plus masker dynamics ranged from 0.25 to 0.75, with the masker having the largest impact when its center frequency was similar to that of the target (*Figure 8f*). When compared at the same high intensity (70 dB SPL), the correlation for animals with hearing loss was typically lower than normal when the masker center frequency was low, especially when the target center frequency was high (*Figure 8g*).

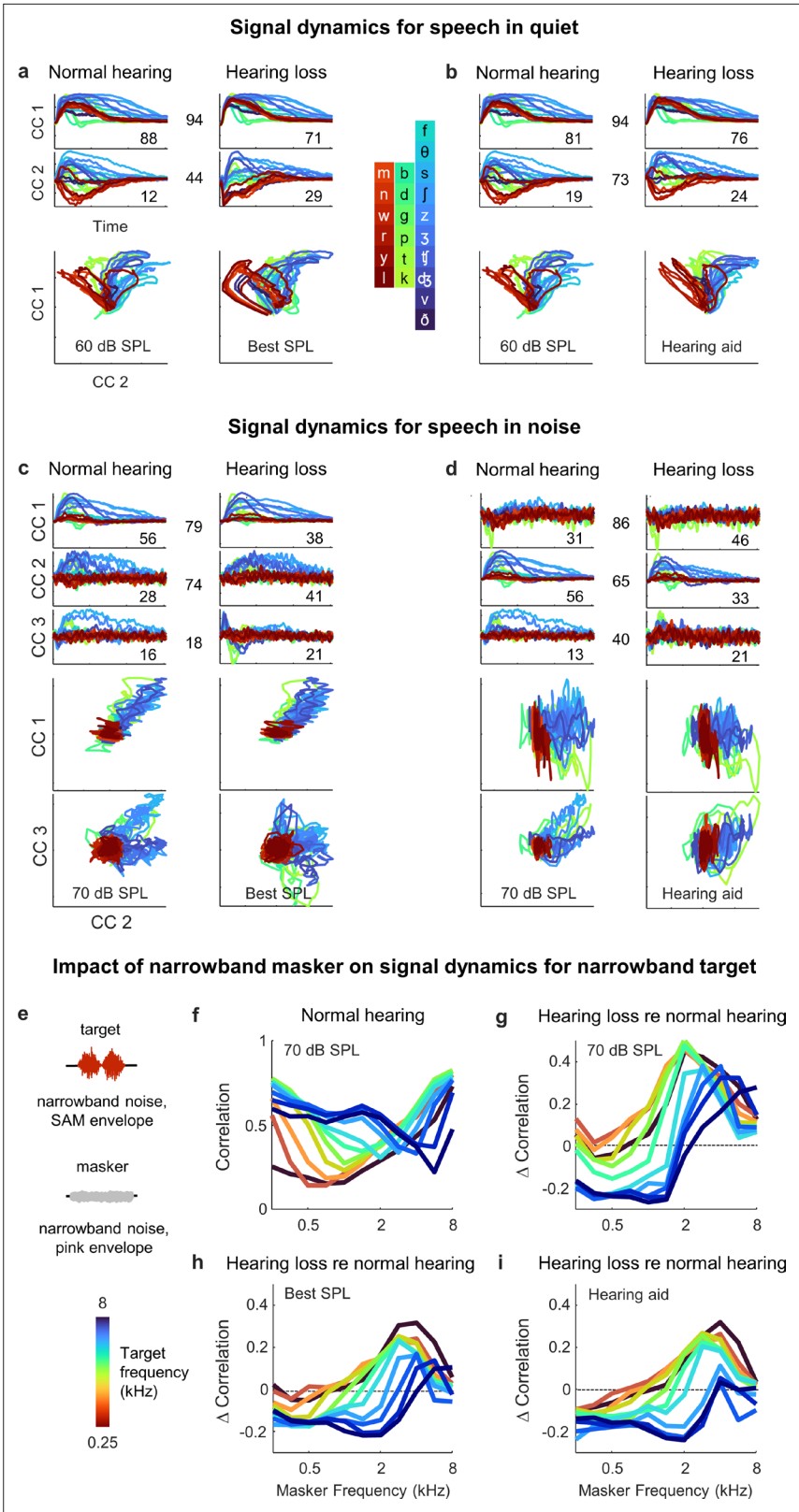

**Figure 8.** Hypersensitivity to background noise after hearing loss. (**a**) Average signal dynamics for speech in quiet at 60 dB SPL for normal hearing animals (left) and at best intensity for animals with hearing loss (right) after joint alignment via multiway canonical correlation analysis (MCCA). The inset value within each panel indicates the average percent of the variance in the bottleneck activations explained by each canonical component (CC). The

*Figure 8 continued on next page*

*Figure 8 continued*

inset value between columns indicates the average correlation between the dynamics after projection onto each pair of CCs. (**b**) Average signal dynamics for speech in quiet at 60 dB SPL for normal hearing animals (left) and with a hearing aid for animals with hearing loss (right) after joint alignment via MCCA. (**c**) Average signal dynamics for speech in noise at 70 dB SPL for normal hearing animals (left) and at best intensity for animals with hearing loss (right) after joint alignment via MCCA. (**d**) Average signal dynamics for speech in noise at 70 dB SPL for normal hearing animals (left) and with a hearing aid for animals with hearing loss (right) after joint alignment via MCCA. (**e**) Schematic diagram showing amplitude modulated narrowband noise target and masker sounds. (**f**) Correlation between bottleneck activations for target noise with and without masker noise at 70 dB SPL. Each line shows the correlation for a different target center frequency as a function of the masker center frequency, averaged across individual animals. (**g**) Change in correlation between bottleneck activations for target noise with and without masker noise at 70 dB SPL for animals with hearing loss relative to animals with normal hearing. (**h**) Change in correlation for animals with hearing loss at best intensity relative to animals with normal hearing at 70 dB SPL. (**i**) Change in correlation for animals with hearing loss with a hearing aid relative to animals with normal hearing at 70 dB SPL.

This regime (high-frequency target, low-frequency masker) is comparable to the speech-in-noise scenario analyzed above (many consonants contain substantial high-frequency content, while speech babble is dominated by low frequencies, see *Figure 7b*), and the increased impact of the masker with hearing loss is consistent with observed hypersensitivity to background noise at the expense of foreground speech. For high-frequency maskers, the correlation for animals with hearing loss was often higher than normal, especially for low-frequency targets. This is unsurprising given the sloping nature of the hearing loss, but this regime (low-frequency target, high-frequency masker) is uncommon in real-world speech.

When the correlation for animals with normal hearing and hearing loss was compared either at the best intensity (*Figure 8h*) or after frequency-weighted amplification with the hearing aid (*Figure 8i*), the pattern of results was similar. In the regime that is most comparable to consonants in speech babble (low-frequency masker, high-frequency target), the correlation for animals with hearing loss was lower than normal. Thus, hearing loss appears to create deficits in spectral processing that manifest as highly nonlinear distortions in cross-frequency interactions that are particularly problematic for real-world speech in noise and cannot be corrected by simply compensating for lost sensitivity.

## Discussion

In this study, we took advantage of recently developed tools for large-scale neural recordings and nonlinear modeling that allowed us to gain new insights into the impact of hearing loss on auditory processing at the network level. We first used a traditional approach to manifold learning to establish that the neural code for speech in the IC can be well described by low-dimensional latent signal dynamics that are common across animals with similar hearing status but fundamentally altered by hearing loss. We then trained a DNN to replicate neural coding in the IC with high accuracy using a framework that also facilitated manifold learning. The DNN exhibited dynamics in response to speech that were similar to those identified directly from IC recordings, and further probing of the DNN dynamics with novel sounds allowed us to identify changes in cross-frequency interactions as a key contributor to the distorted neural coding of speech-in-noise with hearing loss.

### Suprathreshold effects of hearing loss

The effects of hearing loss beyond increased detection thresholds are often ignored in clinical assessment and treatment. But these suprathreshold effects are, in fact, the main problem for many people in real-world settings, such as busy workplaces or social gatherings, where sound intensities are high and amplification via a hearing aid provides little benefit. The clinical neglect of suprathreshold effects is not due to a lack of awareness, but rather to a lack of effective treatments. And the lack of effective treatments stems from a lack of understanding of how the many physiological changes that accompany hearing loss contribute to complex perceptual deficits.

Many specific suprathreshold impairments with plausible links to speech-in-noise perception have been identified, such as decreased frequency selectivity, dynamic range, or temporal resolution (*Moore, 2007*). But the extent to which each of these specific impairments contributes to real-world

hearing problems has been difficult to determine. Our results suggest that changes in spectral processing, particularly in the interactions between different frequencies, are the primary problem. While impaired spectral processing was evident for simple tones as a clustering of dynamical trajectories within the signal manifold, this impairment does not appear to be a major problem for the neural coding of speech per se as suitable amplification corrected many of the distortions caused by hearing loss for speech in quiet. For speech in noise, however, there was a hypersensitivity to background noise with hearing loss that amplification (simple or frequency-weighted) did little to alleviate, and this was consistent with observed interactions between narrowband sounds, which revealed an increased disruption of high-frequency targets by low-frequency maskers both with and without amplification.

These results are consistent with a recent study that found that hearing loss caused the IC activity patterns elicited by different phonemes to become less distinct, and that a hearing aid failed to correct this problem for speech in noise (*Armstrong et al., 2022*). They are also consistent with a body of work demonstrating that listeners with hearing loss struggle to combine temporal envelope cues across frequency channels (*Healy and Bacon, 2002*; *Healy and Carson, 2010*; *Souza and Boike, 2006*; *Grant et al., 2007*). When speech is reduced to a single amplitude-modulated band, speech recognition performance is similar for all listeners, independent of their hearing status, suggesting that temporal processing of the speech envelope per se is unaffected by hearing loss. But as additional amplitude-modulated bands are added, performance increases more for normal hearing listeners than for those with hearing loss, suggesting that the latter group are less able to make use of complementary temporal information across multiple frequency channels. This difference is most pronounced when comparing the ability to make use of temporal modulations in high-frequency channels (4–6 kHz) in the presence of temporal modulations in lower frequency channels (1–2 kHz) (*Grant et al., 2007*), and it does not appear to be a simple consequence of broadened frequency tuning but rather a specific deficit in cross-frequency interactions (*Healy and Carson, 2010*).

## Distorted spectral processing from cochlea to cortex

Understanding exactly what is going wrong with spectral processing after hearing loss at a mechanistic level remains a challenge. The effects of hearing loss on spectral processing in the cochlea have been well described in terms of the observed changes in the frequency tuning curves of individual AN fibers. After hearing loss, the tuning curve 'tip' (corresponding to the characteristic frequency [CF] to which the fiber is most sensitive) becomes less sensitive and may shift toward lower frequencies while the 'tail' (corresponding to frequencies below CF) may become more sensitive (*Young, 2012*). It is difficult to know the degree to which these changes distort the basic tonotopic map in the cochlea (i.e., the relationship between CF and cochlear position) because few studies have identified the cochlear position from which recorded fibers originate. The limited data that exist suggest that the effect of hearing loss on CF tonotopy is modest (*Liberman and Kiang, 1984*), but the effect on the tonotopic map of best frequency (BF; the frequency that elicits the strongest response from a fiber at higher intensities) can be much larger (*Henry et al., 2016*), and can be accompanied by more complex changes in spectral processing such as decreased synchrony capture (*Young, 2012*).

One recent study has provided insight into how the complex spectral distortions in the cochlea impact the neural coding of speech in noise in the AN (*Parida and Heinz, 2022*). In animals with mild-to-moderate sensorineural hearing loss, fibers with high CFs responded excessively to low-frequency sounds and it was this effect (rather than changes in frequency selectivity per se or temporal processing) that appeared to be most responsible for the disrupted neural coding of speech in noise. These results are consistent with our observations of hypersensitivity to background noise and increased disruption of high-frequency targets by low-frequency maskers in the IC. The distortion of spectral processing that we observe as clustering of network dynamics for simple tones in the IC, however, does not appear to be present in the AN. It is difficult to be certain since no directly comparable analysis has been performed on experimental AN responses, but our analysis of simulated AN responses suggests that hearing loss has more complex effects on the signal dynamics in the AN than simply causing clustering (*Figure 4—figure supplement 1*). This transformation from complex distortions in signal dynamics for tones in the AN to the simple clustering observed in the IC could arise from normal central processing of distorted peripheral inputs, but it is also possible that plastic changes in central processing that follow hearing loss facilitate a shift into a new dynamical regime.

Distorted spectral processing has also recently been observed in the auditory cortex after mild-to-moderate sloping sensorineural hearing loss (*McGill et al., 2022*). Animals displayed behavioral hypersensitivity for detection of tones at the edge frequencies around which the hearing loss increased from mild to moderate as well as an overrepresentation of these frequencies in the cortical tonotopic map of BF. The mechanisms underlying these phenomena are not entirely clear. Some cortical neurons tuned to these edge frequencies exhibited increased neural gain and synchrony, and direct stimulation of thalamocortical afferents demonstrated that hearing loss caused an increase in gain within the local cortical circuit. But the frequency tuning of the stimulated afferents was unknown and, thus, it is difficult to separate the effects that were cortical in origin from those that were inherited from lower levels. It is possible that the altered neural representation of spectral features that we observed in the IC results in changes in the coactivation patterns across the cortical network, prompting the plastic reorganization in the cortex. Future research should be focused on developing a coherent model of how peripheral and central changes combine to create auditory processing deficits, perhaps through coordinated experiments across many brain areas in a single species.

## A new focus for hearing aid design

The complex suprathreshold effects of hearing loss that are evident in the distorted neural signal dynamics observed in this study present a difficult challenge for hearing aid designers. Current hearing aids compensate for changes in the threshold and dynamic range of auditory processing using a framework built around a bank of bandpass filters with automatic gain control. The signal processing that can be performed by such a framework is highly constrained, and it is difficult to imagine how it could be used to compensate for problems such as hypersensitivity to background noise that involve highly nonlinear interactions across frequency bands. It is possible that with a better understanding of exactly how different frequencies are interacting, new signal processing frameworks can be designed to offset the more complex effects of hearing loss. But engineering such a framework that is flexible enough to provide benefit in a wide range of real-world settings will require conceptual advances that may not be forthcoming in the near term.

One alternative approach to improving the perception of speech in noise that is already showing promise is speech enhancement, that is, the suppression of background noise. Hearing aids have offered noise suppression for many years, but the simple filters based on low-order statistics that are used by current devices often provide little real-world benefit (*Brons et al., 2014*; *Cox et al., 2014*). Recent research on speech enhancement via DNNs has demonstrated the potential to yield dramatic improvements in performance (*Wang, 2017*; *Luo and Mesgarani, 2019*). This 'deep denoising' may prove effective in situations with a single talker where it is obvious which sound is of interest. But in others, for example, with multiple talkers, a sound that is of primary interest one minute may become a distraction the next. It may be possible to implement cognitive control to direct enhancement toward the sound of interest but there are many significant technical challenges that must be overcome before this approach can be widely applied (*Geirnaert et al., 2021*).

A more flexible alternative is to identify optimal processing algorithms for hearing aids empirically by providing DNNs with the data they need to learn how best to transform sounds in order to elicit normal neural activity from an impaired auditory system (*Lesica, 2018*; *Drakopoulos and Verhulst, 2022*). By taking advantage of the nonlinear capacity of DNNs with minimal assumptions, it should be possible to identify novel general-purpose algorithms that go well beyond the hand-designed processing in current devices. Such algorithms would be especially valuable in important contexts such as listening to music – a major problem for hearing aid users (*Madsen and Moore, 2014*) – in which denoising cannot help. There are, of course, limits to the degree of hearing restoration that any hearing aid can provide in cases of severe hearing loss. But the vast majority of people with hearing loss have only mild-to-moderate cochlear damage (*Wilson et al., 2017*), and there should be sufficient functionality remaining within the auditory system for a hearing aid to leverage when attempting elicit the necessary patterns of neural activity.

## Modeling biological neural networks with DNNs

Building computational models of sensory processing has been a long-standing goal in systems neuroscience. Current models of the sensory periphery can be highly accurate. For example, there are numerous models of the cochlea that faithfully capture the transformation of incoming sound

into basilar membrane motion and AN activity (*Saremi et al., 2016*; *Verhulst et al., 2018*). Models of sensory processing in the brain, however, have generally been much less accurate, with even the best models missing out on a significant fraction of the explainable variance in subcortical and cortical neural activity (*Williamson et al., 2016*; *Rahman et al., 2020*; *McFarland et al., 2013*; *Vintch et al., 2015*).

Until recently, efforts to model central sensory processing were constrained by the difficulties of fitting high-capacity models with limited experimental data. But deep learning has provided a new approach to fitting models that are well matched to sensory processing, and new methods for large-scale electrophysiology can provide the required big data. Initial efforts to use DNNs to model the sensory periphery have shown that they can be as accurate as hand-designed biophysical models. In one recent study, a DNN trained to replicate an established model of the cochlea provided accurate and generalizable predictions of the model's response to speech and tones (*Baby et al., 2021*). In another study, a DNN trained on retinal ganglion cell activity predicted nearly all of the explainable variance in responses to natural images (*Maheswaranathan et al., 2019*).

DNN models of sensory processing in the brain have also been shown to outperform traditional models. DNN models of V1 responses to natural images explained 50 and 80% of the explainable variance in single-unit spike counts and calcium activity, respectively (*Cadena et al., 2019*; *Walker et al., 2019*), while DNN models of V4 explained 90% of the explainable variance in multi-unit spike counts (*Bashivan et al., 2019*). DNN models of A1 responses to speech and other natural sounds perform similarly well, explaining much of the explainable variance in high-gamma activity, fMRI voxel responses, or time-varying spike rates (*Keshishian et al., 2020*; *Kell et al., 2018*; *Pennington and David, 2022*).

Our results improve on these initial successes in several important ways. Firstly, our models simulate neural activity with full temporal resolution, that is, spike times with millisecond precision. While lower temporal resolution may be sufficient to describe sensory processing in some contexts, precise spike timing carries critical information about speech (*Garcia-Lazaro et al., 2013*). Secondly, the activity produced by our models is nearly indistinguishable from that recorded experimentally, capturing more than 95% of the explainable variance in many cases. This is especially remarkable considering the full temporal resolution (with lower resolution, variance at fine time scales, which is typically the most difficult to capture, is simply ignored). Finally, our use of a low-dimensional bottleneck allows us to achieve this performance within a framework that also provides a compact and interpretable description of the latent dynamics that underlie the full network activity patterns.

With these advances, it should now be possible to use computational models of the brain for exploratory basic research, with benefits that are both scientific (studies are no longer data limited) and ethical (animal experiments can be limited to confirmatory studies), as well as for technology development (such as improved hearing aids, as described above). With further effort, it may be possible to build models that are not only black-box simulators but also mirror the underlying structure of biological systems (*Jazayeri and Ostojic, 2021*; *Chung and Abbott, 2021*). Such models would provide powerful platforms for testing mechanistic hypotheses and developing new ways to address complex network-level dysfunctions that remain difficult to treat (such as tinnitus).

## Methods
### Experimental protocol

Experiments were performed on 12 young-adult gerbils of both sexes that were born and raised in standard laboratory conditions. Six of the animals were exposed to noise when they were 16–18 weeks old. (These six were chosen from among many that were noise exposed based on the pattern of hearing loss that they exhibited: sloping mild-to-moderate in both ears.) The number of animals used was not predetermined. Because of the investigative nature of the study, the key outcome measures were not known in advance and, thus, a pre-study power analysis based on anticipated effect sizes was not possible. The duration of the data collection from each animal was predetermined based on the results of preliminary experiments in which the amount of neural activity required for manifold analysis and deep learning to yield stable results was assessed. Assignment to the control and hearing loss groups was random on a per-animal basis (i.e., animals from the same litter were often assigned to different groups). Investigators were not blinded during data collection or analysis (since the difference

between animals with normal hearing and hearing loss is immediately apparent upon the observation of sound-evoked neural activity), but all analyses were fully automated and objective. ABR recordings and large-scale IC recordings were made from all animals when they were 20–24 weeks old. All experimental protocols were approved by the UK Home Office (PPL P56840C21).

## Noise exposure

Mild-to-moderate sensorineural hearing loss was induced by exposing anesthetized gerbils to high-pass filtered noise with a 3 dB/octave roll-off below 2 kHz at 118 dB SPL for 3 hr (*Armstrong et al., 2022*; *Suberman et al., 2011*). For anesthesia, an initial injection of 0.2 ml per 100 g body weight was given with fentanyl (0.05 mg per ml), medetomidine (1 mg per ml), and midazolam (5 mg per ml) in a ratio of 4:1:10. A supplemental injection of approximately 1/3 of the initial dose was given after 90 min. Internal temperature was monitored and maintained at 38.7°C.

## Preparation for large-scale IC recordings

Animals were placed in a sound-attenuated chamber and anesthetized for surgery with an initial injection of 1 ml per 100 g body weight of ketamine (100 mg per ml), xylazine (20 mg per ml), and saline in a ratio of 5:1:19. The same solution was infused continuously during recording at a rate of approximately 2.2 µl per min. Internal temperature was monitored and maintained at 38.7°C. A small metal rod was mounted on the skull and used to secure the head of the animal in a stereotaxic device. The pinnae were removed and speakers (Etymotic ER-2) coupled to tubes were inserted into both ear canals along with microphones (Etymotic ER-10B+) for calibration. The frequency response of these speakers measured at the entrance of the ear canal was flat (±5 dB) between 0.2 and 8 kHz. Two craniotomies were made along with incisions in the dura mater, and a 256-channel multi-electrode array was inserted into the central nucleus of the IC in each hemisphere (*Armstrong et al., 2022*). The arrays were custom-designed to maximize coverage of the portion of the gerbil IC that is sensitive to the frequencies that are present in speech.

## Auditory brainstem responses

Before beginning the IC recordings, ABRs were measured. Subdermal needles were used as electrodes with the active electrodes placed behind the ear over the bulla (one on each side), the reference placed over the nose, and the ground placed in a rear leg. Recordings were bandpass-filtered between 300 and 3000 Hz. The parallel ABR method (*Polonenko and Maddox, 2019*) was used, with randomly timed tones at multiple frequencies presented simultaneously and independently to each ear. The tone frequencies were 500, 1000, 2000, 4000, and 8000 Hz. Each tone was five cycles long and multiplied by a Blackman window of the same duration. Tones were presented at a rate of 40 per s per frequency with alternating polarity for 100 s at each intensity. The activity recorded in the 30 ms following each tone was extracted and thresholds for each frequency were defined as the lowest intensity at which the root mean square (RMS) of the median response across presentations was more than twice the RMS of the median activity recorded in the absence of sound.

## Sounds presented during IC recordings

### Speech

Sentences were taken from the TIMIT corpus (*Garofolo, 1993*) that contains speech read by a wide range of American English speakers. The entire corpus excluding 'SA' sentences was used (approximately 4.5 hr) and split into training and test sets (4.25 hr and 0.25 hr, respectively; not to be confused with the suggested training/test subdivisions in the TIMIT documentation). The training set was presented twice, once on its own and once with background noise. The test set was presented four times, twice in quiet and twice with the same background noise. The intensity for each sentence was chosen at random from 55, 65, 75, or 85 dB SPL. The speech-to-noise ratio (SNR) was chosen at random from either 0 or 10 when the speech intensity was 55 or 65 dB SPL (as is typical of a quiet setting such as a home or an office) or –10 or 0 when the speech intensity was 75 or 85 dB SPL (as is typical of a noisy setting such as a pub). The intensity of the sentences for the two presentations of the test set in quiet were identical, as were the intensity, SNR, and specific noise used for the two presentations of the test set with background noise.

### Noise

Background noise sounds were taken from the Microsoft Scalable Noisy Speech Dataset (**Reddy et al., 2019**), which includes recordings of environmental sounds from a large number of different settings (e.g., café, office, roadside) and specific noises (e.g., washer-dryer, copy machine, public address announcements). A total of 4.5 hr of unique noises were used to match the duration of the presented speech. The intensity of the noise presented with each sentence was determined by the intensity of the speech and the SNR as described above.

## Multi-unit activity

MUA was measured from recordings on each channel of the electrode array as follows: (1) a bandpass filter was applied with cutoff frequencies of 700 and 5000 Hz; (2) the standard deviation of the background noise in the bandpass-filtered signal was estimated as the median absolute deviation/0.6745 (this estimate is more robust to outlier values, e.g., neural spikes, than direct calculation); (3) times at which the bandpass-filtered signal made a positive crossing of a threshold of 3.5 standard deviations were identified and grouped into bins with a width of 1.3 ms. Only units with a signal correlation (across repeated trials of the speech in the test set) of 0.2 or higher were used for manifold learning and DNN training (420 ± 24 [mean ± SD] units from each animal out of 512 total channels).

## Analysis of recorded neural activity

For each animal, the MUA was represented as an $M \times T$ matrix, where $M$ is the number of units and $T$ is the number of time bins. Separate matrices $\boldsymbol{R}_{train}$, $\boldsymbol{R}_{test1}$, and $\boldsymbol{R}_{test2}$ were formed for the training set and each repetition of the test set (see 'Speech' above).

## Dimensionality of signal manifold

We applied PCA to $\boldsymbol{R}_{train}$ (after subtracting the mean from each row) to obtain the PCs, ranked in order of the amount of neural variance they explain. We projected the activity in $\boldsymbol{R}_{test1}$ onto a chosen number of PCs to obtain the latent dynamics within the manifold spanned by those PCs, yielding a new $D \times T$ matrix $\boldsymbol{X}_{test1} = \boldsymbol{Z}\boldsymbol{R}_{test1}$, where $\boldsymbol{Z}$ is the $D \times M$ matrix containing the first $D$ PCs. We reconstructed the activity in $\boldsymbol{R}_{test1}$ from the latent dynamics as $\hat{\boldsymbol{R}}_{test1} = \boldsymbol{Z}^T\boldsymbol{X}_{test1}$ (plus the originally subtracted means) and measured the total variance explained as the ratio of the covariance between $\boldsymbol{R}_{test1}$ and $\hat{\boldsymbol{R}}_{test1}$ and the square root of the product of their variances. We reconstructed the activity in $\boldsymbol{R}_{test2}$ from the same latent dynamics as $\hat{\boldsymbol{R}}_{test2} = \boldsymbol{Z}^T\boldsymbol{X}_{test1}$ (plus the means of the rows of $\boldsymbol{R}_{test2}$) and measured the signal variance explained as the ratio of the covariance between $\boldsymbol{R}_{test2}$ and $\hat{\boldsymbol{R}}_{test2}$ and the square root of the product of their variances. We defined the dimensionality of the signal manifold for each animal based on the number PCs required to explain 95% of the signal variance.

## Similarity of signal dynamics

We measured the similarity between the signal dynamics for different animals as the variance explained after linear regression of one set of dynamics $\boldsymbol{X}_{test1}$ onto another $\boldsymbol{Y}_{test1} = \boldsymbol{X}_{test1}\beta + \varepsilon$, where $\beta$ is a matrix of regression coefficients and $\varepsilon$ is a vector of error terms.

## Deep neural network models

DNNs were used to transform sound input into neural activity across four stages: (1) a SincNet layer (**Ravanelli and Bengio, 2018**) with 48 bandpass filters of length 32 samples, each with two learnable parameters (center frequency, bandwidth), followed by symmetric log activations $Y = \text{sgn}(x) \log(|x| + 1)$; (2) a stack of five 1-D convolutional layers, each with 128 filters of length 32 samples and stride 2, followed by PReLU activations; (3) a 1-D bottleneck convolutional layer with a specified number of filters of length 32 and stride 1, followed by PReLU activations; and (4) a linear readout layer followed by exponential activations. The only hyperparameter that was varied was the number of filters in the bottleneck layer. For comparison with a linear–nonlinear (LN) model, we used a network with the same stages 1 and 4 and a single convolutional layer between them with 128 filters of length 256 samples and stride 1, followed by PReLU activations.

## Training

Models were trained to transform 24,414.0625 kHz sound input frames of length 8192 samples into 762.9395 Hz neural activity frames of length 192 samples (corresponding to temporal decimation by

a factor of 5 via the strided convolutions in the encoder block plus a final cropping layer that removed 32 samples at the start and end of each frame to eliminate convolutional edge effects). Sound inputs were scaled such that an RMS of 1 corresponded to a level of 94 dB SPL. Training was performed in MATLAB on a local PC with GPUs (2x NVIDIA RTX 3080) with a batch size of 64 for 10 epochs and took about 8 hr for a typical model. The Adam optimizer was used with a learning rate of 0.0001. The optimization was framed as Poisson regression with loss function $\sum_{M,T} \left( \hat{R} - R \log \left( \hat{R} \right) \right)$ **,** where $R$ is the recorded neural activity, $\hat{R}$ is the network output, $M$ is the number of units, and $T$ is the number of time bins.

## Validation

For each animal, data were split into training and test sets (see 'Speech' above). The training set was used to learn the optimal values of the DNN parameters. The final performance of the optimized network was measured on the test set by calculating the percent of the explainable variance in the recorded responses that was explained by the network outputs based on the ratio of the covariance between $R_{test1}$ and $\hat{R}$ and the covariance between $R_{test1}$ and $R_{test2}$ , where $\hat{R}$ is the network output and $R_{test1}$ and $R_{test2}$ are the recorded responses to the two presentations of the test speech.

## Analysis of bottleneck activations

For each animal, the activations in the bottleneck layer for all sounds from a given class (e.g., all pure tones or all consonants in noise) were extracted to form the $D_b$ x $T$ signal dynamics matrix $X$, where $D_b$ is the number of bottleneck channels and $T$ is the number of time bins. For visualization, we applied PCA to the dynamics in $X$ and projected them onto a chosen number of PCs.

## Sounds presented to trained DNNs

### Pure tones

100 ms tones with frequencies ranging from 500 Hz to 8000 Hz in 0.2 octave steps; intensities ranging from 25 dB SPL to 100 dB SPL in 5 dB steps; 10 ms cosine on and off ramps; and a 100 ms pause between tones.

### SAM noise, fixed modulation depth

100 ms bursts of bandpass noise with cutoff frequencies of 500 and 8000 Hz; a sinusoidal envelope with frequencies ranging from 10 Hz to 240 Hz in 10 Hz steps and a modulation depth of 1; intensities ranging from 25 dB SPL to 100 dB SPL in 5 dB steps; 10 ms cosine on and off ramps; and a 100 ms pause between tones.

### SAM noise, fixed modulation frequency

100 ms bursts of bandpass noise with cutoff frequencies of 500 and 8000 Hz; a sinusoidal envelope with a modulation depth ranging from 0.1 to 1 in 20 logarithmic steps and a frequency of 30 Hz; intensities ranging from 25 dB SPL to 100 dB SPL in 5 dB steps; 10 ms cosine on and off ramps; and a 100 ms pause between tones.

### Isolated consonants

Speech utterances were taken from the Articulation Index LSCP (LDC Cat# LDC2015S12). Utterances were from 10 American English speakers (five males, five females). Each speaker pronounced consonant-vowel syllables made from all possible combinations of 22 consonants and 13 vowels. For each utterance, the border between the consonant and vowel was identified in a semi-automated manner (a clustering algorithm [MATLAB *linkage*] was applied to the spectrogram time bins to identify two clusters based on a correlation metric and the border between them was inspected and corrected if needed), values after the end of the consonant were set to zero (with a 2 ms linear ramp), and the utterance was truncated to 200 ms. Utterances were presented in random order with a 175 ms pause between sounds at intensities ranging from 25 dB SPL to 100 dB SPL in 5 dB steps.

### Multi-talker speech babble noise

Continuous speech from 16 different British English speakers from the UCL Scribe database (https://www.phon.ucl.ac.uk/resource/scribe) was summed to create speech babble. The intensity of the

babble was set based on the intensity of the isolated consonants to achieve a speech-to-noise ratio of 3 dB.

### Narrowband target

100 ms bursts of bandpass noise with center frequencies ranging from 500 Hz to 8000 Hz in 0.5 octave steps and a bandwidth of 0.5 octaves; a sinusoidal envelope with a modulation depth of 1 and a frequency of 20 Hz; intensities ranging from 25 dB SPL to 100 dB SPL in 5 dB steps; 10 ms cosine on and off ramps; and a 100 ms pause between tones.

### Narrowband noise

100 ms bursts of bandpass noise with center frequencies ranging from 500 Hz to 8000 Hz in 0.5 octave steps and a bandwidth of 0.5 octaves; a pink noise envelope (power scaled as inverse of frequency) with a modulation depth ((peak – trough)/peak) of 1; an intensity matched to that of the narrowband target; 10 ms cosine on and off ramps; and a 100 ms pause between tones.

### Hearing aid simulation

A 10-channel wide-dynamic range compression hearing aid was simulated using a program provided by Prof. Johsua Alexander (Purdue University) (*Alexander and Masterson, 2015*). The crossover frequencies between channels were 200, 500, 1000, 1750, 2750, 4000, 5500, 7000, and 8500 Hz. The intensity thresholds below which amplification was linear for each channel were 45, 43, 40, 38, 35, 33, 28, 30, 36, and 44 dB SPL. The attack and release times (the time constants of the changes in gain following an increase or decrease in the intensity of the incoming sound, respectively) for all channels were 5 and 40 ms, respectively. The gain and compression ratio for each channel were fit individually for each ear of each gerbil using the Cam2B.v2 software provided by Prof. Brian Moore (Cambridge University) (*Moore et al., 2010*). The gain before compression typically ranged from 10 dB at low frequencies to 30 dB at high frequencies. The compression ratios typically ranged from 1 to 2.5, that is, the increase in sound intensity required to elicit a 1 dB increase in the hearing output ranged from 1 dB to 2.5 dB when compression was engaged.

### Representational similarity analysis

For each animal, the signal dynamics matrix $X$ was reshaped to yield $\widetilde{X}$, an $S \; x \; (D_b \; x \; T_S)$ matrix, where $S$ is the number of sounds from a given class and $T_S$ is the number of times bins associated with an individual sound. An $S \; x \; S$ representational dissimilarity matrix (RDM) was formed, with each entry equal to one minus the correlation between a pair of rows in $\widetilde{X}$. To compute overall representational similarity, the upper triangular values (excluding the diagonal) from two $\widetilde{X}$ matrices were reshaped into vectors and the correlation between them was computed. For speech, dynamics were averaged across all instances of each consonant before RDMs were computed.

### Canonical correlation analysis

To align two sets of signal dynamics, $U = XA$ and $V = YB$ were computed using QR factorization and singular value decomposition (MATLAB *cannoncorr*), where $X$ and $Y$ are the matrices containing the original dynamics, $A$ and $B$ are the matrices containing the canonical components, and $U$ and $V$ are the aligned dynamics. Overall similarity after alignment was computed as $\sum_{d=1}^{D_b} \rho \left( U_d, V_d \right) * \left( \rho \left( \hat{X}_d, X \right)^2 + \rho \left( \hat{Y}_d, Y \right)^2 \right) / 2$, with the second term in the product acting as the 'weight for the correlation associated with each pair of CCs' that is referred to in the 'Results.' $U_d$ and $V_d$ are the projections of $X$ and $Y$ onto the $d^{th}$ pair of canonical components, $\hat{X}_d = X * \left( a_d * \left( a_d^T a_d \right)^{-1} * a_d \right)$ and $\hat{Y}_d = Y * \left( b_d * \left( b_d^T b_d \right)^{-1} * b_d \right)$ are the reconstructions of $X$ and $Y$ from the $d^{th}$ pair of canonical components and $\rho$ denotes point-by-point correlation. To jointly align

more than two sets of dynamics, multiway CCA was used (*de Cheveigné et al., 2019*) (MATLAB NoiseTools *nt_mcca*).

### Decoding signal dynamics

For each animal, the signal dynamics matrix $X$ was reshaped such that each row contained the dynamics for one consonant instance. A support vector machine was trained (MATLAB *fitcecoc*) to identify consonants from signal dynamics with a max-wins voting strategy based on all possible combinations of binary classifiers and tenfold cross-validation.

## Acknowledgements

The authors thank A de Cheveigné, S Ostojic, JÁ Gallego, and F Bruford for their advice. This work was supported by a Wellcome Trust Senior Research Fellowship (200942/Z/16/Z) and a grant from the UK Engineering and Physical Sciences Research Council (EP/W004275/1). The funding sources were not involved in study design, data collection and interpretation, or the decision to submit the work for publication.

## Additional information

### Competing interests

Nicholas A Lesica: is a co-founder of Perceptual Technologies. The other authors declare that no competing interests exist.

### Funding

| Funder | Grant reference number | Author |
| --- | --- | --- |
| Wellcome Trust | 200942/Z/16/Z | Shievanie Sabesan<br>Nicholas A Lesica |
| Engineering and Physical Sciences Research Council | EP/W004275/1 | Ciaran Bench<br>Fotios Drakopoulos<br>Nicholas A Lesica |

The funders had no role in study design, data collection and interpretation, or the decision to submit the work for publication. For the purpose of Open Access, the authors have applied a CC BY public copyright license to any Author Accepted Manuscript version arising from this submission.

### Author contributions

Shievanie Sabesan, Investigation, Writing – review and editing; Andreas Fragner, Conceptualization, Software, Supervision, Methodology, Writing – review and editing; Ciaran Bench, Fotios Drakopoulos, Software, Investigation, Methodology; Nicholas A Lesica, Conceptualization, Software, Supervision, Funding acquisition, Investigation, Methodology, Writing – original draft, Writing – review and editing

### Author ORCIDs

Nicholas A Lesica (ID) http://orcid.org/0000-0001-5238-4462

### Ethics

All experimental protocols were approved by the UK Home Office (PPL P56840C21). Every effort was made to minimize suffering.

### Decision letter and Author response

Decision letter https://doi.org/10.7554/eLife.85108.sa1
Author response https://doi.org/10.7554/eLife.85108.sa2

## Additional files

### Supplementary files
• MDAR checklist

### Data availability
The metadata, ABR recordings, and a subset of the IC recordings analyzed in this study are available on figshare (DOI:10.6084/m9.figshare.845654). We have made only a subset of the IC recordings available because they are also being used for commercial purposes. These purposes (to develop improved assistive listening technologies) are distinct from the purpose for which the recordings are used in this manuscript (to better understand the fundamentals of hearing loss). Researchers seeking access to the full set of neural recordings for research purposes should contact the corresponding author via e-mail to set up a material transfer agreement. The custom code used for training the deep neural network models for this study is available at https://github.com/nicklesica/dnn, (copy archived at *Lesica, 2023*).

The following dataset was generated:

| Author(s) | Year | Dataset title | Dataset URL | Database and Identifier |
|-----------|------|---------------|-------------|-------------------------|
| Lesica N | 2023 | Data from Sabesan et al., 2023 | https://doi.org/10.6084/m9.figshare.845654 | figshare, 10.6084/m9.figshare.845654 |

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
