## [Editor Report]

This fundamental work uses deep neural networks to simulate activity evoked by a wide range of stimuli and demonstrates systematic differences in latent population representations between hearing-impaired and normal-hearing animals that are consistent with impaired representations of speech in noise. The evidence supporting the conclusions is compelling, and the neural-network approach is novel with potential future applications. The research will be of interest to auditory neuroscientists and computational scientists.

---

## [Decision Letter]

**Decision letter after peer review:**

Thank you for submitting your article "Distorted neural signal dynamics create hypersensitivity to background noise after hearing loss" for consideration by *eLife*. Your article has been reviewed by 2 peer reviewers, and the evaluation has been overseen by a Reviewing Editor and Barbara Shinn-Cunningham as the Senior Editor. The following individual involved in review of your submission has agreed to reveal their identity: Stephen V David (Reviewer #2).

Essential revisions:

1) The reviewers raised concerns about the generalizability of the study's approach. The approach hinges on the deep neural network getting things right, such that it generalizes across sounds. The study's claims would be substantially more convincing if the authors included data that validates their model's predictions about coding of speech in noise, tones in noise, and SAM noise in NH vs. HL animals. It appears that they have data in hand for speech and speech-in-noise from the same animals that they could analyze using methods already in the manuscript. If they are unable to validate any of these predictions, the authors should revise the manuscript to emphasize that they remain predictions until they can be validated with additional data in a different study.

2) Several labs have studied changes in inferior colliculus and cortex, but their work is not acknowledged in this manuscript. For example, the work by the Sanes lab at NYU and Polley lab at Harvard have advanced theories around decreased inhibition to accommodate reduced peripheral input. This work has also implicated deficits in temporal processing that do not at the surface appear consistent with the current study (e.g., see Sanes and Yao *eLife* 2018). The authors would want to place their work in the context of these and other works more clearly.

3) Details about some statistical tests were hard to find (e.g., only in Table S1), but it also appears that the authors still make important statements without statistical justification, for example, related to NH/HL+noise vs. NH/HL+quiet (Figures 6 vs. Figures 7). There are several other cases where conclusions, e.g., about dimensionality, are not supported by a statistical test. The authors would want to make sure that all their conclusions are supported quantitatively.

4) The authors would also want to flesh out the argument for why the same effects would not be present in the nerve.

*Reviewer #1 (Recommendations for the authors):*

Suggestions for authors:

-at the end of intro, I think you could make it a little more explicit that the DNN is being trained to predict PC responses from sound.

For people who will not be familiar with the practical constraints that necessitate a design with separate groups of normal-hearing and hearing-impaired gerbils, you might state explicitly early on that you are comparing separate groups of NH and HI gerbils.

Figure 1 caption – why are there two sample sizes given in the last sentence?

Line 83 – you might give some flavor as to the noise types that were used

Figure 2A is really well done -- you made a pretty complicated list of comparisons quite straightforward to follow

Line 111 and Figure 2 – I must be missing something, but I don't see how you can approach 100% explained signal variance given the way that I think it is calculated. Doesn't the noise variance show up in the denominator?

Line 130 – I suggest motivating/justifying the additional linear transformation for the reader

Figure 3 – can anything be said about how the left panel of Figure 3d looks pretty different from Figure 2i?

I don't understand what constitutes a "recorded unit". The methods refer to multi-unit activity. Is a unit just an electrode from the array of 512? Or is there spike sorting being performed? How is any non-stationarity in the recordings dealt with (e.g. if neurons die, or the brain moves a little w.r.t. the electrode array)?

Line 177 – I found myself wondering how the results would compare to a more conventional model with a STRF for each neuron. I suspect lots of people will wonder how the DNN compares to something like that.

Lines 183-185 – give numbers for the similarity, to be parallel to earlier sections

Lines 219-220 – the "clustering of dynamics" referred to here was not all that evident to this reviewer from eyeballing the figure – please make what you mean more explicit, and clarify how this is different from refs 18 and 19

Lines 232-233 – I recommend making the RDMs more conventional and just having more of them in the figure – I think people will find the asymmetry confusing when they page through the paper

Line 241 – are the numbers mean and SD? Please specify.

Line 246 – I didn't completely understand what would constitute a distortion to the "overall structure of the dynamics" – could you give an example?

Multiple figures – I don't think the asterisks are defined clearly, and I believe the mean different things in different figures. Please label more explicitly, and/or mention in each caption.

Line 287 – I found myself wondering about the possible effect of phase shifts or increases in response latency, which one might imagine could occur with hearing loss. I think the analysis would be highly vulnerable to this, especially given that the encoding of modulation is partly synchrony-based. The fact that the modulation analysis shows pretty similar results for NH and HI suggests there is not much of this, but some readers may wonder about this.

At several points throughout the paper, I found myself wondering about the effects of compression. I would have been greatly interested to see an analysis that separately manipulated compression (e.g., turning it off), to see how much benefit it produces on restoring normal-like responses. I also would have liked to see some discussion of its effects.

Line 344 – for this analysis, I was hoping to have chance levels explicitly stated, and ideally labeled on the graph.

Figure 7d – this panel is pretty confusing, partly because the SPL numbers are inside particular plots, so it is not completely clear what they apply to, and partly because the little numbers in the plots are not labeled or defined anywhere.

Line 404-406 – how does this jive with the findings of distorted tonotopy from the Heinz lab?

Line 408 and onwards – the text refers to CCs but the figure is labeled as PCs

Line 438 – why is this coherence rather than correlation?

Line 457-459 – these lines state the conclusions of the paper, but I think they could be more explicitly linked to the results that come earlier. Explain why the distortions are nonlinear, and explain why the effects involve cross-frequency interactions.

Lines 472-473 – the statement here (and the earlier one on line 33) seems a little too strong given the widespread prevalence of noise reduction, and the widespread use of speech in noise diagnostics in audiometric evaluations

Line 484 and earlier – can the clustering be explained merely by audibility (e.g., all the stimuli that are inaudible cluster together, for the uninteresting reason that they do not evoke a response)?

Line 496 – the claim here needs a reference

Line 511 – I wanted to know more about the absence of evidence of clustering in nerve responses. This seems critical.

Line 586 and onwards – I think the conclusions/suggestions here should be tempered given that there are almost surely going to be limits to how well DNN models trained in this way will generalize to arbitrary stimuli. And you might acknowledge some of these limitations.

Line 606 – I think it might be helpful to specify what a material transfer agreement would involve – does this just mean someone agrees not to share it with anyone else?

Line 691 – why is "/ 0.6745" here? Is this a typo?

Line 697 – what is a "unit"?

Line 768 – I wondered whether setting values to 0 without any windowing might induce artifacts…

Line 784 – it seems plausible that the hearing aid settings are suboptimal. In particular, the extent of compression is based on some informal optimization in humans. Could this partly explain the less than complete restoration of normal responses?

Line 810 – it would help to link this to the weights that are described in the Results section. It took me a couple reads to make the connection.

Overall, the statistical tests and quantitative comparisons are somewhat buried. There are a lot of statistical comparisons via color map (i.e., Figures 2H-I and 3D) where a scatter or bar plot with error bars might be more helpful.

*Reviewer #2 (Recommendations for the authors):*

1. While there are many open questions around central deficits following hearing loss, several labs have studied changes in IC and cortex, but their work is not acknowledged in this manuscript. In particular the Sanes lab at NYU and Polley lab at Harvard have advanced theories around decreased inhibitory tone to accommodate diminished bottom-up drive. Relevant to the current study, this work has implicated deficits in temporal processing that do not at the surface appear consistent with the current study (eg, see Sanes and Yao *eLife* 2018). Hearing loss and the neural coding of sound are complex, so the concern is not about the validity of the current results as much as how they fit in with the existing literature. Currently, the manuscript reads as if this previous work was never completed, and that issue should be addressed.

2. In general, the results of fairly sophisticated analyses are presented clearly, which is great. After some hunting, it was possible to find important details about some statistics in Table S1, but it appears that the authors still make important statements without statistical justification. Of particular importance to the main conclusions, the increased dissimilarity for NH/HL+noise vs. NH/HL+quiet (Figure 6 vs. Figure 7) needs to be demonstrated by a quantitative comparison between them. Table S1 doesn't appear to contain anything about comparisons between data in the different figures. Please provide quantitative support for the statement that "… neither was sufficient to bring the similarity close to normal levels" (Line 379). There are several other cases where conclusions, eg, about dimensionality, are not supported by a statistical test. The authors should make sure that all their conclusions are supported quantitatively. It would also

3. The performance of the DNN is impressive, providing a reasonable motivation for the subsequent analysis of "bottleneck PCs" for activity simulated by the model. However, one worries that since the models were not fit to stimuli tested in the simulation, that the results may not actually be reciprocated in actual neural activity. One contrast, in particular (speech in quiet vs. speech in noise), was actually collected experimentally, and it seems like the authors could validate their decoding analysis with the actual neural data. Can't the neural responses be projected back into the bottleneck space and be used to decode the same way as the DNN simulations? Such an analysis would substantially strengthen the study. Alternatively, the authors should include a caveat in the Discussion that the DNN simulations may not actually generalize to actual neural activity. The authors may wish to argue that this is a small concern, but the finding of such low-dimensional PC bottleneck is quite surprising, and it's not clear if dimensionality would be as small if the actual stimuli (pure tones, SAM noise) were included in the fit set.

---

## [Author Response]

Essential revisions:1) The reviewers raised concerns about the generalizability of the study's approach. The approach hinges on the deep neural network getting things right, such that it generalizes across sounds. The study's claims would be substantially more convincing if the authors included data that validates their model's predictions about coding of speech in noise, tones in noise, and SAM noise in NH vs. HL animals. It appears that they have data in hand for speech and speech-in-noise from the same animals that they could analyze using methods already in the manuscript. If they are unable to validate any of these predictions, the authors should revise the manuscript to emphasize that they remain predictions until they can be validated with additional data in a different study.

We understand the concern about generalization to out-of-sample inputs. A model trained on one set of inputs may not necessarily produce accurate simulations of responses to a different set of inputs. Thus, in order for our analysis of model latent representations of non-speech sounds to be compelling, we must demonstrate that the model responses to such sounds are an accurate simulation of the true responses.

(Some of the reviewer comments seem to suggest that we also need to demonstrate the model’s accuracy for speech and speech-in-noise sounds, but this is already shown in the original Figure 3c; the performance shown is for speech and speech-in-noise sounds that were not part of the training set.)

The non-speech sounds that we used in the study were pure tones and SAM noise. For pure tones, we were able to verify that the models generalized well by using recordings from the same animals on which the models were originally trained (without including the tone responses in training). For SAM noise, we did not have recordings from the original animals. However, this provided an opportunity to further test the generality of the model by using transfer learning to predict responses from new animals for which we had responses to SAM noise as well as a small sample of speech. We froze the DNN encoder after training on animals from the original dataset and retrained only the linear readout using the speech responses for the new animals. We then tested the ability of the updated models to predict SAM noise responses for the new animals, and they performed well. This new work is described in the revised Results (see new Figure 3f,g and associated text).

The one class of sounds for which we were not explicitly able to validate the models is hearing aid-amplified speech. While these sounds are not qualitatively different from standard speech and we have no particular reason to believe the model predictions for these sounds would be inaccurate, we have added a note to the text to indicate the lack of validation.

2) Several labs have studied changes in inferior colliculus and cortex, but their work is not acknowledged in this manuscript. For example, the work by the Sanes lab at NYU and Polley lab at Harvard have advanced theories around decreased inhibition to accommodate reduced peripheral input. This work has also implicated deficits in temporal processing that do not at the surface appear consistent with the current study (e.g., see Sanes and Yao eLife 2018). The authors would want to place their work in the context of these and other works more clearly.

We have tried to do more to place our results within the context other related studies. The study from the Polley lab that is most closely related to ours is McGill et al. (2022) in which they study the downstream effects of mild-to-moderate sensorineural hearing loss. (Other studies from the Polley lab use a model of extreme neuropathy, which is too different from mild-to-moderate sensorineural hearing loss to allow for meaningful comparisons.) The key findings of McGill et al. that relate to our study are (1) that hearing loss induces behavioral hypersensitivity for detection of tones at the frequencies around which the hearing loss increases from mild to moderate; (2) that these frequencies are overrepresented after rearrangement of the cortical tonotopic map; and (3) that (some) cortical neurons located in this region of the tonotopic map exhibit increased gain and synchrony in their responses.

Our work does not investigate the circuit-level mechanisms that underlie the observed effects of hearing loss (e.g., bottom-up drive vs. local E-I balance). The work from the Sanes lab is focused on these mechanisms and it is difficult for us to see how further consideration of our results in conjunction with theirs can lead to additional insights. The specific study suggested by Reviewer 2, Yao and Sanes (2018) is focused on developmental hearing loss, which makes it even more difficult to compare with our work. Also, the reviewer suggests that their results are somehow inconsistent with ours, but they are not. Their abstract states “We found that developmental HL … did not alter brainstem temporal processing.” Our results also suggest that HL does not alter brainstem temporal processing, and this is consistent with many other studies that have found that HL does not impact temporal processing in the early auditory pathway (see Parida and Heinz (2022) for another recent example). Understanding how temporal processing deficits arise at the level of the cortex after hearing loss is not something that our work can help with; for that we must continue to look to the Sanes lab and others who are focused on such questions.

What we can do is try to synthesize our results with others related to mild-to-moderate sensorineural hearing loss from the auditory nerve and cortex in order to better understand the transformation that takes place along the way. We have added a new section to the Discussion “Distorted spectral processing from cochlea to cortex” along these lines. Perhaps the most salient point we can take from this exercise is the recognition that coordinated studies are needed to develop a coherent picture.

3) Details about some statistical tests were hard to find (e.g., only in Table S1), but it also appears that the authors still make important statements without statistical justification, for example, related to NH/HL+noise vs. NH/HL+quiet (Figures 6 vs. Figures 7). There are several other cases where conclusions, e.g., about dimensionality, are not supported by a statistical test. The authors would want to make sure that all their conclusions are supported quantitatively.

We have added statistical tests to support our assertion that the distortions in signal dynamics caused by hearing loss are more pronounced for speech in noise than for speech in quiet (Figures 6 and 7). Whether compared at best intensity or after amplification with a hearing aid, and whether measured via RSA or CCA, the distortions were much smaller for speech in quiet than for speech in noise, with all differences highly significant (the largest p-value was less than 1e-6). We have included this information in the revised Results.

We have also added distribution plots and statistical tests to support our assertion that the signal dynamics differ between pairs of animals with normal hearing and hearing loss more than between pairs of animals with the same hearing status (new Figures 2i and 3e). Whether based on the signal manifold as identified via PCA or via DNN, signal dynamics were much more similar for pairs of animals with the same hearing status than for pairs of animals with different hearing status.

Reviewer 2 also suggested statistical tests in two instances where we did not make explicit comparisons between groups because we did not feel that these comparisons would be informative. But we include the statistical tests here for completeness:

Figure 2c,g. The dimensionality of the signal manifold

We assert only that the signal manifold is low dimensional with both normal hearing and hearing loss. We have added the range of values for the dimensionality of the signal manifold for each group of animals to the revised text. A t-test indicates that the average dimensionality with hearing loss is significantly lower than with normal hearing in the statistical sense (p = 0.04, mean = 4.8 for HL and 6.8 for NH). But since we cannot say whether or not this difference is significant in the functional sense in and of itself (as opposed to the many detailed differences in the signal manifold with and without hearing loss that we go on to analyze in the rest of the study), we did not include it in the Results.

Figure 3c. The predictive power of the DNN model

We assert only that the model performs well for both normal hearing and hearing loss. In fact, the predictive power was generally higher for hearing loss than for normal hearing: separate t-tests for each bottleneck dimensionality (see Figure 3c) yielded p-vales of 0.08, 4e-4, 1e-3, 3e-3, 1e-3, and 1e-3; all but the first of these indicates significantly better performance for hearing loss even after (Bonferroni correction). But, inasmuch as we do not follow up on this difference to understand how it arises, we don’t think it is appropriate to include it in the Results.

4) The authors would also want to flesh out the argument for why the same effects would not be present in the nerve.

Broadly speaking, many of the effects that we see (e.g., hypersensitivity to background noise) are present in the AN. What does not appear to be present in the AN is the specific form of distorted spectral processing that we observe in the IC as a clustering of signal dynamical trajectories within the latent representation of the DNN model.

Providing a definitive answer to the question of why these effects are present in the IC and not the AN is beyond us (though we added some speculative ideas to the new section in the revised Discussion). We can, however, provide more explicit evidence that the distortions in spectral processing in the AN and the IC are, in fact, different. To do this, we simulated AN responses to pure tone sounds with and without hearing loss, as suggested by Reviewer 1, and performed the same analyses of the signal dynamics as we did for the IC.

We found no evidence in the AN of the clustering of dynamical trajectories that is present in the IC. (In fact, the effects of hearing loss on spectral processing in the AN as revealed through this analysis appear to be much more complex than in the IC). We also demonstrated that the clustering of dynamical trajectories that we observed in the latent representation of the IC DNN model was also evident in experimental IC responses. These new analyses are described in Figure 4 —figure supplement 1 of the revised manuscript.

Reviewer #1 (Recommendations for the authors):Suggestions for authors:Figure 1 caption – why are there two sample sizes given in the last sentence?

It is just one (large) sample size: n = 544,362.

Line 83 – you might give some flavor as to the noise types that were used

This information is in the Methods.

Line 111 and Figure 2 – I must be missing something, but I don't see how you can approach 100% explained signal variance given the way that I think it is calculated. Doesn't the noise variance show up in the denominator?

With the full complement of PCs, all of the variance in a dataset can be fully explained. So, in general, it should be no surprise that a subset of that variance can also be fully explained; in fact, it would be impossible for this not to be the case. It is, however, potentially surprising that so much of the signal variance can be explained with so few PCs.

I don't understand what constitutes a "recorded unit". The methods refer to multi-unit activity. Is a unit just an electrode from the array of 512? Or is there spike sorting being performed? How is any non-stationarity in the recordings dealt with (e.g. if neurons die, or the brain moves a little w.r.t. the electrode array)?

Spike sorting was not performed, but the extraction of multi-unit activity involved some processing. This is described in the Methods. Non-stationarity was ignored.

Line 287 – I found myself wondering about the possible effect of phase shifts or increases in response latency, which one might imagine could occur with hearing loss. I think the analysis would be highly vulnerable to this, especially given that the encoding of modulation is partly synchrony-based. The fact that the modulation analysis shows pretty similar results for NH and HI suggests there is not much of this, but some readers may wonder about this.

RSA is insensitive to phase shifts; CCA is not. This difference is part of the motivation for using both methods rather than just one or the other.

At several points throughout the paper, I found myself wondering about the effects of compression. I would have been greatly interested to see an analysis that separately manipulated compression (e.g., turning it off), to see how much benefit it produces on restoring normal-like responses. I also would have liked to see some discussion of its effects.

We did an extensive analysis of the effects of compression on the neural coding of speech in a previous paper (Armstrong et al., Nat Biomed Eng., 2022).

Figure 7d – this panel is pretty confusing, partly because the SPL numbers are inside particular plots, so it is not completely clear what they apply to, and partly because the little numbers in the plots are not labeled or defined anywhere.

We understand that this figure can be confusing, but the formatting and labelling are exactly the same as in all of the previous figures, e.g., 4d and 5d. We tried several other designs for this figure, but none were judged to be better.

Line 404-406 – how does this jive with the findings of distorted tonotopy from the Heinz lab?

The relationship between our findings and the distorted tonotopy that has been observed in the auditory nerve is considered in detail in the Discussion. To our knowledge, the potential for (frequency-weighted) amplification to mitigate the effects of distorted tonotopy on speech coding at the level of the auditory nerve has not been tested.

Line 438 – why is this coherence rather than correlation?

As used here, they are equal. We have relabeled as correlation since that is likely to be more familiar to readers.

Line 484 and earlier – can the clustering be explained merely by audibility (e.g., all the stimuli that are inaudible cluster together, for the uninteresting reason that they do not evoke a response)?

That would be possible, but (1) there is also clustering of responses to low-frequency tones that evoke strong responses and (2) the high-frequency tones do, in fact, elicit a response (see Figure 4d).

Line 496 – the claim here needs a reference

The relevant papers are cited throughout the paragraph.

Line 691 – why is "/ 0.6745" here? Is this a typo?

No. That is the scaling factor required to transform an estimate of mean absolute deviation into an estimate of standard deviation.

(See https://en.wikipedia.org/wiki/Median_absolute_deviation)

Line 697 – what is a "unit"?

The MUA-processed signal from one recording channel.

Line 784 – it seems plausible that the hearing aid settings are suboptimal. In particular, the extent of compression is based on some informal optimization in humans. Could this partly explain the less than complete restoration of normal responses?

We agree that it is plausible that the hearing aid settings are suboptimal. But we think it is unlikely that this suboptimal fitting is the main reason why hearing aids are unable to restore neural responses to normal. We did an extensive analysis of the effects of hearing aids on the neural coding of speech in a previous paper (Armstrong et al., Nat Biomed Eng., 2022).